# Neuronal ER-plasma membrane junctions couple excitation to Ca²⁺-activated PKA signaling

Nicholas C. Vierra [1] ✉, Luisa Ribeiro-Silva[1], Michael Kirmiz[1], Deborah van der List[1], Pradeep Bhandari[2], Olivia A. Mack[3], James Carroll[3], Elodie Le Monnier[2], Sue A. Aicher[3], Ryuichi Shigemoto [2] & James S. Trimmer [1] ✉

Junctions between the endoplasmic reticulum (ER) and the plasma membrane (PM) are specialized membrane contacts ubiquitous in eukaryotic cells. Concentration of intracellular signaling machinery near ER-PM junctions allows these domains to serve critical roles in lipid and Ca²⁺ signaling and homeostasis. Subcellular compartmentalization of protein kinase A (PKA) signaling also regulates essential cellular functions, however, no specific association between PKA and ER-PM junctional domains is known. Here, we show that in brain neurons type I PKA is directed to Kv2.1 channel-dependent ER-PM junctional domains via SPHKAP, a type I PKA-specific anchoring protein. SPHKAP association with type I PKA regulatory subunit RI and ER-resident VAP proteins results in the concentration of type I PKA between stacked ER cisternae associated with ER-PM junctions. This ER-associated PKA signalosome enables reciprocal regulation between PKA and Ca²⁺ signaling machinery to support Ca²⁺ influx and excitation-transcription coupling. These data reveal that neuronal ER-PM junctions support a receptor-independent form of PKA signaling driven by membrane depolarization and intracellular Ca²⁺, allowing conversion of information encoded in electrical signals into biochemical changes universally recognized throughout the cell.

Membrane contacts between ER and PM (ER-PM junctions) are present in all eukaryotic cells and serve as sites for lipid and Ca²⁺ signaling and homeostasis[1,2]. ER cisternae juxtaposed with the PM in brain neurons were among the first ER-PM junctions described[3] and are frequently associated with successive parallel cisternae of ER stacked immediately adjacent to the PM-associated ER cistern. Although these ER-PM junctional domains—comprising ER-PM junctions and their associated ER stacks (also called subsurface cisternae)—are abundant in the somata of brain neurons and were first described over 60 years ago[4], their neurophysiological functions have remained unclear. Moreover,

specific molecules recruited to these domains in brain neurons and mechanisms that recruit and organize these molecules are mostly unknown.

A set of key proteins mediates protein–lipid and protein–protein interactions that form ER-PM junctions[1]. Among these is the voltage-gated potassium channel Kv2.1, which is present at most ER-PM junctions in hippocampal and cortical pyramidal neurons[5]. This PM-embedded protein has a cytoplasmic C-terminus that interacts with ER-resident VAP proteins[6–8]. By tethering ER to the PM, Kv2.1 is unique among ion channels in that it functions as both an ion-conducting

[1]Department of Physiology and Membrane Biology, University of California Davis School of Medicine, Davis, CA, USA. [2]Institute of Science and Technology Austria (ISTA), Klosterneuburg, Austria. [3]Chemical Physiology and Biochemistry Department, Oregon Health & Science University, Portland, OR, USA. ✉e-mail: ncvierra@ucdavis.edu; jtrimmer@ucdavis.edu

channel and a structural organizer of ER-PM junctions[7,9,10]. Furthermore, Kv2.1 creates compartmentalized $Ca^{2+}$ signaling domains in the soma by recruiting L-type voltage-gated $Ca^{2+}$ channels (LTCCs) to ER-PM junctions, enhancing LTCC spatial and functional association with ER ryanodine receptor (RyR) $Ca^{2+}$ release channels at these sites and supporting excitation-transcription coupling[9,11].

Signaling events mediated by PKA, a key effector of the second messenger cyclic AMP (cAMP), play critical roles in regulating diverse aspects of neuronal function and plasticity. Because PKA can be activated by various receptor-based pathways and phosphorylate numerous substrates, its spatial compartmentalization by A-kinase anchoring proteins (AKAPs) is essential for defining its specific physiological functions[12]. The PKA holoenzyme comprises two catalytic (PKA-C) subunits and two cAMP-binding and AKAP-interacting regulatory (R) subunits, either RI or RII, which respectively generate type I and type II PKA. Certain neuronal AKAPs target type II PKA to dendritic spines[13], where it receives localized input from neurotransmitter receptor-coupled signaling pathways to trigger synaptic potentiation and spine structural plasticity. Unlike synaptic type II PKA signaling systems, no specific mechanism for compartmentalizing PKA activity is known for the soma, a neuronal compartment that must integrate diverse neuronal inputs while lacking the membrane-delimited compartmentalization of biochemical reactions inherent to dendrites and axons. Moreover, in contrast to type II PKA, the long-standing model is that soma-enriched type I PKA[14] is localized throughout the cytosol, with little known of its specific anchoring mechanisms or neurophysiological functions. Here, we identify a type I PKA signalosome tightly associated with stacked ER cisternae at Kv2.1-mediated ER-PM contacts that couples membrane depolarization to PKA activity. These results reveal a functional association of PKA with neuronal ER-PM junctional domains and define a mode of compartmentalized type I PKA signaling.

## Results

### SPHKAP clusters type I PKA near Kv2.1-associated ER-PM junctions

To identify molecules enriched within the neuronal ER-PM junction nano-environment, we immunopurified Kv2.1-containing protein complexes from DSP-crosslinked mouse brain homogenates and analyzed their protein constituents by tandem mass spectrometry. As described previously[6,9] we removed from our dataset any proteins immunopurified by anti-Kv2.1 antibodies from DSP-crosslinked mouse brain homogenates prepared from Kv2.1 knockout mice. We also immunopurified protein complexes associated with the axonal Kv1.2 $K^+$ channel to further assess the specificity and sensitivity of our Kv2.1 immunopurification protocol. In the set of proteins specific to immunopurified Kv2.1-containing protein complexes, the most abundant protein detected after Kv2.1 was SPHKAP[15,16] (Table 1), a type I PKA-specific AKAP[13]. Consistent with SPHKAP's exclusive anchoring of type I PKA[15,16], we detected an abundance of RI and PKA-C, but not RII, in immunopurified Kv2.1 complexes (Table 1). These proteins were not present in parallel anti-Kv2.1 immunopurifications from Kv2.1 knockout mice, or in immunopurified Kv1.2-containing proteins complexes, the latter of which contained many known Kv1.2 associated proteins (Supplementary Table 2). These results suggested that SPHKAP and type I PKA are in close proximity to ER-PM junction-enriched Kv2.1 channels in brain neurons.

To determine the cellular and subcellular distribution of SPHKAP in the brain, we next generated poly- and mono-clonal antibodies against SPHKAP. These antibodies revealed robust SPHKAP immunolabeling in hippocampal, cortical, cerebellar, striatal, and hypothalamic neurons in mouse brain sections (Fig. 1a). Relative to other brain regions, RI has been reported to be particularly abundant in the cerebellum[17], where we found SPHKAP labeling to be especially prominent. Like Kv2.1[18], SPHKAP displayed restricted localization to the

## Table 1 | PKA-related proteins specifically copurifying with Kv2.1

| Rank (abundance) | Protein ID | UniProt accession | Mean | s.e.m. |
|---|---|---|---|---|
| 1 | Kv2.1 | Q03717 | 100.0 | NA |
| **2** | **SPHKAP** | **E9PUC4** | **32.2** | **1.0** |
| 3 | Kv2.2 | A6H8H5 | 31.6 | 0.5 |
| 5 | VAPA | Q9WV55 | 25.3 | 1.7 |
| **8** | **PKA RIα** | **Q9DBC7** | **15.3** | **1.5** |
| **9** | **PKA RIβ** | **P12849** | **12.9** | **0.6** |
| 15 | VAPB | Q8BH80 | 7.6 | 1.4 |
| 16 | Junctophilin-3 | Q9ET77 | 6.3 | 1.3 |
| **17** | **PKA Cβ** | **P68181** | **5.6** | **0.3** |
| **21** | **PKA RIα** | **A2AI69** | **5.2** | **1.2** |
| **27** | **PKA Cα** | **P05132** | **4.7** | **0.2** |

Table of PKA signaling proteins (bold) copurifying with mouse brain Kv2.1. Values are spectral counts normalized to Kv2.1 (three biological replicates).

soma and proximal dendrites. Indeed, multiplex labeling of brain sections for Kv2.1, SPHKAP, and PKA-RI revealed an overall similarity in the distribution of all three proteins on neuronal somata and proximal dendrites in the hippocampal formation (Supplementary Fig. 1a, b). SPHKAP and RI exhibited a highly punctate distribution in hippocampal pyramidal cell somata (Fig. 1b, c) and in cultured hippocampal neurons (Supplementary Fig. 2a, b), with immunolabeling for these proteins concentrated in clusters up to 1 μm in diameter. The punctate somatodendritic RI immunolabeling we observed was consistent with the previous observation of the somatodendritic enrichment of RIβ clusters in the pyramidal cell layers of hippocampal areas CA1-3[14]. Comparison of Kv2.1 and SPHKAP immunolabeling with that of VGAT, a marker of GABAergic terminals that form synapses prominent on neuronal somata, revealed co-clustering of SPHKAP with Kv2.1, but little overlap of SPHKAP with VGAT or GAD67, another protein enriched in GABAergic terminals, further supporting the somatic localization of SPHKAP (Supplementary Fig. 3a, b).

To determine whether ER-PM junction-forming Kv2 channels influence neuronal SPHKAP clustering, we compared SPHKAP immunolabeling in brain sections from control wild-type (WT) and knockout (KO) mice lacking Kv2.1 (Kv2.1 KO[19]), Kv2.2 (Kv2.2 KO[20]), or both Kv2 channels (DKO[21]). Both Kv2.1 and Kv2.2 channels possess a phospho-FFAT motif in their cytoplasmic C-terminus, allowing them to form and cluster at ER-PM junctions via a physical interaction with VAP proteins[7]. Loss of either or both Kv2 channels reduced SPHKAP cluster size (Fig. 1d, e), indicating that Kv2 channels are necessary for large clusters of SPHKAP in brain neurons. Moreover, loss of Kv2.1 reduced the spatial association of RI immunolabeling with that of SPHKAP in CA1 neurons (Fig. 1f, g). While impacting SPHKAP and RI co-clustering, eliminating Kv2 expression did not affect the overall immunolabeling intensity for either SPHKAP or RI (Fig. 1h), suggesting that Kv2 channels are a determinant of SPHKAP-RI subcellular distribution but not their abundance. The positive correlation between Kv2.1, SPHKAP, and RI immunolabeling (Fig. 1i) across numerous neurons further suggested that ER-PM junction-forming Kv2 channels promote the assembly of SPHKAP and type I PKA at these sites.

Like Kv2.1 and Kv2.2[6,8,22], SPHKAP possesses a phospho-FFAT motif[23] predicted to mediate binding to VAP proteins, which act as ER scaffolds for diverse proteins including those forming ER-organelle membrane contacts[24]. Accordingly, we found that VAP immunolabeling density was enriched at SPHKAP clusters (Supplementary Fig. 3c, d), suggesting that SPHKAP may recruit VAP to specific ER subdomains. We further explored and quantified the spatial relationship between immunolabeled SPHKAP, Kv2.1, RI, and VAP proteins in

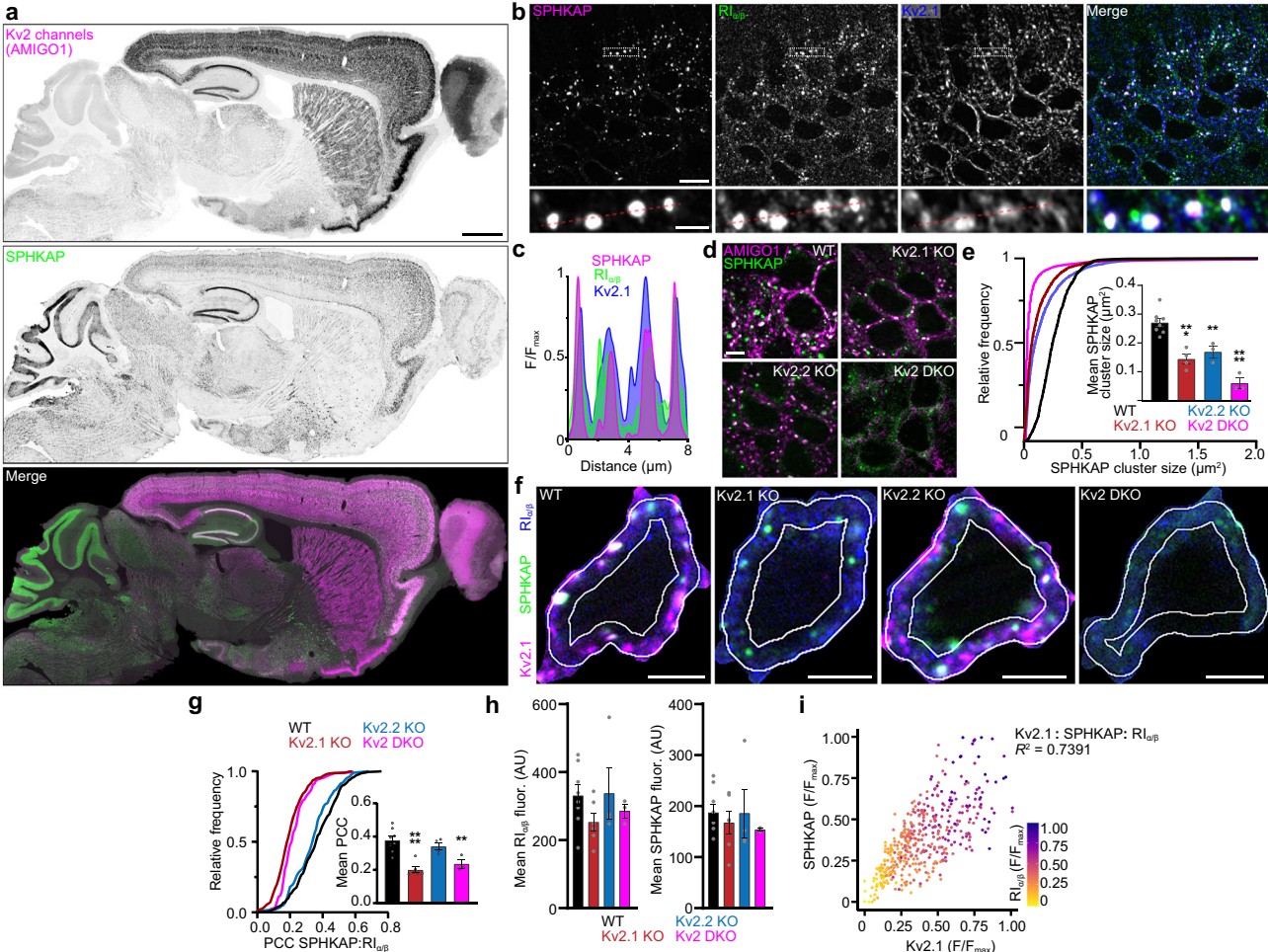

**Fig. 1 | SPHKAP clusters type I PKA near Kv2.1-associated ER-PM junctions.**
**a** Mouse brain section immunolabeled for the Kv2 channel subunit AMIGO-1 (magenta) and SPHKAP (green) (representative of $n = 8$ mice). Scale bar, 1 mm. **b**, **c** SPHKAP (magenta), PKA-RIα/β (green), and Kv2.1 (blue) immunolabeling in hippocampal area CA1 in mouse brain (**b**). Scale bar, 10 μm. Lower panels, expanded views of regions marked by boxes (representative of $n = 8$ mice). Scale bar, 2 μm. Red line indicates region selected for intensity profile line scan (**c**). **d** Representative images of AMIGO-1 and SPHKAP immunolabeling in hippocampal area CA1 in brain sections from WT ($n = 8$), Kv2.1 KO ($n = 4$), Kv2.2 KO ($n = 3$), and Kv2 DKO ($n = 3$) mice. Scale bar, 5 μm. **e** Mean ± s.e.m. SPHKAP cluster size obtained in **d**. Each point represents 1 mouse. Statistical significance determined by one-way ANOVA followed by Dunnett's multiple comparisons test (vs. WT); ****$P < 0.0001$, ***$P = 0.0003$, **$P = 0.0055$. **f** Representative images of hippocampal area CA1 neuron somata from WT, Kv2.1 KO, Kv2.2 KO, and Kv2 DKO brain sections

immunolabeled for Kv2.1 (magenta), SPHKAP (green), and RI (blue). Region of interest (ROI) in which immunofluorescence was quantified shown in white (scale bar, 5 μm). **g** Frequency distribution and mean ± s.e.m. SPHKAP:RI PCC values in CA1 neurons from WT ($n = 8$), Kv2.1 KO ($n = 6$), Kv2.2 KO ($n = 3$), and Kv2 DKO ($n = 3$) mice. Each point represents 1 mouse; statistical significance determined using one-way ANOVA followed by Tukey's multiple comparisons test (vs. WT); ****$P < 0.0001$, **$P = 0.0076$. **h** Mean ± s.e.m. immunofluorescence signal intensity of RI (left) and SPHKAP (right) in CA1 pyramidal neurons. Each point represents 1 mouse (as in **g**); statistical significance assessed using one-way ANOVA followed by Tukey's multiple comparisons test (vs. WT); no significant differences were found. **i** Plot of normalized Kv2.1, SPHKAP, and RI (color gradient) immunofluorescence intensity in CA1 pyramidal neurons from $n = 4$ mice. Each point represents 1 cell ($n = 429$ cells). All source data are provided as a Source Data file.

---

cultured hippocampal neurons using super-resolution ground state depletion (GSD) microscopy in total internal reflection fluorescence (TIRF) mode, allowing detection of proteins associated with or near the PM with a lateral resolution of 20–40 nm. From the reconstructed Kv2.1 and SPHKAP super-resolution maps we determined that 19.4 ± 2.4% of Kv2.1 pixels overlapped with SPHKAP immunolabeling. Intensity profile line scans and measurement of centroid nearest-neighbor distances from reconstructed GSD maps further revealed a close spatial association of SPHKAP with Kv2.1, RI, and VAPs (Fig. 2a–f).

We complemented this super-resolution microscopy with proximity ligation assay (PLA), an immunoassay that enables detection of protein pairs within 40 nm of each other[25]. We quantified PLA-reported Kv2.1-SPHKAP (Fig. 2g) and SPHKAP-VAPA/B (Fig. 2h) associations in cultured hippocampal neurons including those in which SPHKAP expression was reduced by shRNA-mediated knockdown (KD)

(Supplementary Fig. 4). We observed abundant PLA signal between SPHKAP and both Kv2.1 and VAP proteins in neurons expressing a control scrambled shRNA, but not in neurons expressing distinct anti-SPHKAP shRNAs or in control neurons in which one antibody of the PLA pair was excluded, further supporting a close association between SPHKAP, Kv2.1, and VAPs.

Conventional light and GSD super-resolution microscopy consistently demonstrated large clusters (up to ~1 μm in diameter) of SPHKAP immunolabeling in neuronal somata. To determine the SPHKAP-associated cellular structure, we next performed immuno-electron microscopy (EM) in rat and mouse brain sections. We observed dense clusters of SPHKAP immunoperoxidase reaction product and immunogold particles on and between stacked ER cisternae adjacent to PM in CA1 pyramidal neurons in rat (Fig. 3a) and mouse (Fig. 3b, c and Supplementary Fig. 5) brain sections. We found SPHKAP

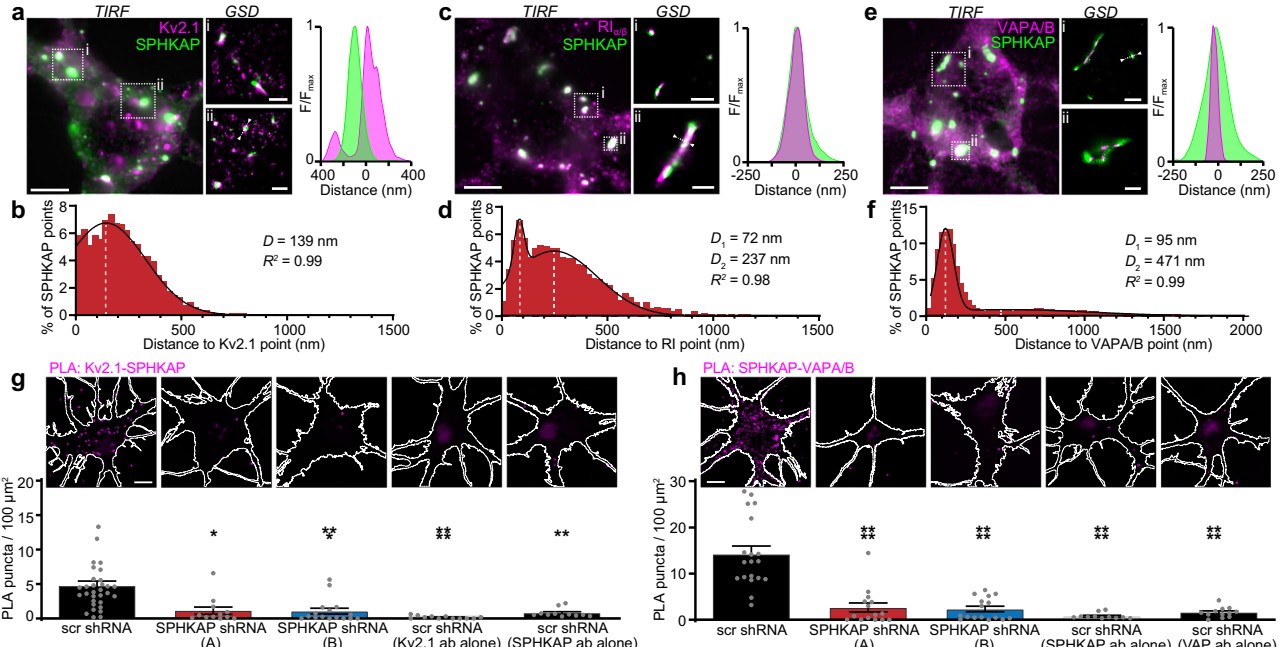

**Fig. 2 | SPHKAP-type I PKA closely associates with Kv2.1 and VAPs.** Total internal reflection fluorescence (TIRF) images and ground-state depletion (GSD) reconstruction maps of cultured neurons immunolabeled for SPHKAP and **a** Kv2.1, **c** RI, or **e** VAPA/B (TIRF scale bars, 5 μm); representative of *n* = 5 [Kv2.1], *n* = 7 [RI], and *n* = 5 [VAPA/B] cells. Two magnified regions marked by boxes are shown for each panel; x-y fluorescence intensity profile from region indicated by dotted line show areas of proximity between SPHKAP and **a** Kv2.1 (scale bars *i* & *ii* = 1 μm), **c** RI (scale bar *i* = 1 μm; scale bar *ii* = 500 nm), and **e** VAPA/B (scale bar *i* = 1 μm; scale bar *ii* = 500 nm). **b**, **d**, **f** Histograms of nearest neighbor distances to SPHKAP centroids for **b** Kv2.1 (*n* = 4863 particles/3 cells), **d** RI (*n* = 3972 particles/4 cells), or **f** VAPA/B (*n* = 7140 particles/3 cells) centroids. Histograms were fit with a **b** Gaussian or **d, f** sum of two Gaussians function; mean values of distributions and $R^2$ values depicted on graphs. **g** Representative images of proximity ligation assay (PLA)

puncta (magenta) in neurons expressing scrambled shRNA (scr) or one of two distinct anti-SPHKAP shRNAs (A or B) and immunolabeled for Kv2.1 and SPHKAP. The neuron periphery is outlined in each image; scale bar, 10 μm. Mean ± s.e.m PLA signal density is presented below each image. Each point represents 1 cell; *n* = 31 (scr), 12 (A), 16 (B), 12 (Kv2.1 ab alone), and 11 (SPHKAP ab alone) cells; statistical significance determined using one-way ANOVA followed by Tukey's multiple comparisons test (vs. scr); ****$P$ < 0.0001, ***$P$ = 0.0001, **$P$ = 0.0004, *$P$ = 0.0009. **h** As in **g**, but for SPHKAP-VAPA/B PLA signal density. Each point represents 1 cell; *n* = 19 (scr), 15 (A), 16 (B), 11 (SPHKAP ab alone), and 12 (VAP ab alone) cells; statistical significance determined using one-way ANOVA followed by Tukey's multiple comparisons test (vs. scr); ****$P$ < 0.0001. Source data are provided as a Source Data file.

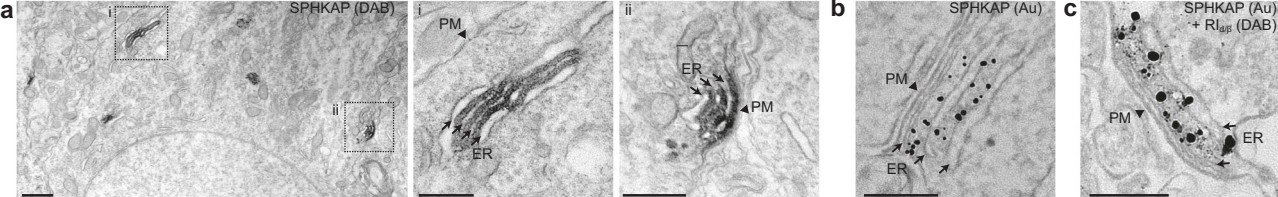

**Fig. 3 | SPHKAP-type I PKA localizes to stacked ER cisternae at ER-PM junctions.** **a** EM images of SPHKAP-immunoperoxidase DAB reaction product acquired from the soma of a CA1 pyramidal neuron in a rat brain section. High-magnification images of insets are provided on right. Scale bars, 1 μm (left panel) and 500 nm (insets i & ii). Images are representative of results obtained from *n* = 2 rats. **b** EM image of SPHKAP-immunogold particles acquired from the soma of a CA1

pyramidal neuron in a mouse brain section. Scale bar, 200 nm. Images are representative of results obtained from *n* = 2 mice. **c** EM image of SPHKAP-immunogold particles and RI-immunoperoxidase reaction product acquired from the soma of a CA1 pyramidal neuron in a mouse brain section. Scale bar, 200 nm. Images are representative of results obtained from *n* = 2 mice.

---

immunoperoxidase immunoreactivity (SPHKAP-IPIR) on subsurface cisternae making direct contact with the PM (25 of 53 cisternae: 47%) as well as stacked ER cisternae deeper in the cytoplasm (28 of 53 cisternae; 53%), consistent with the original description of stacked ER cisternae tightly associated with the PM and also found closer to the nucleus in brain neurons[4]. SPHKAP-IPIR structures were not detected at presynaptic sites or in astrocytic glial processes.

Similarly, SPHKAP immunogold labeling was enriched in the cytosolic space between stacked ER cisternae, with 116.9 ± 9.9 immunogold particles/μm² on stacked ER cisternae as compared to only 1.7 ± 0.2 immunogold particles/μm² in the cytoplasm ($P$ = 2.5 × 10⁻¹⁹, Student's *t* test, *n* = 40 cells; 539 particles [cisternae] and 509 particles

[cytoplasm] total), highlighting SPHKAP's robust association with stacked ER. Double SPHKAP/RI immuno-EM labeling showed RI's colocalization with SPHKAP on stacked ER (Fig. 3c and Supplementary Fig. 5d). Although SPHKAP immunoreactivity was strongly associated with stacked ER, SPHKAP immunogold labeling was not detected in the narrow gap between the PM and the ER (i.e., the ER-PM junction itself) in EM micrographs. We suspect that this result may be a technical artifact related to the strong fixation conditions required for EM, which when combined with the narrower gap between the PM and ER (10–15 nm) as compared to the space between subsequent stacked cisternae (20–30 nm)[26] could preclude antibody access to epitopes in this constrained space. Despite this, our ability to detect PLA signal

between Kv2.1's cytoplasmic C-terminus and SPHKAP (Fig. 2g) provides experimental evidence supporting SPHKAP's presence on the PM-facing ER, as the ER membrane would be expected to present a barrier to PLA signal generation between PM Kv2.1 and SPHKAP localized between stacked ER. Together, these results support that SPHKAP-type I PKA complexes are an enriched component of the electron-dense material intercalating between neuronal subsurface cisternae[4] and indicate that like SPHKAP, type I PKA is concentrated on stacked ER cisternae associated with Kv2.1-containing ER-PM junctions.

## RI anchoring drives SPHKAP-RI co-clustering

We consistently observed large co-clusters of endogenously expressed SPHKAP and RI in neurons in brain sections and in culture. SPHKAP, unlike most AKAPs, has two RI binding sites and can thus anchor two type I PKA holoenzymes[16], each of which contains two R subunits, suggesting a potential mechanism for SPHKAP-RI cluster formation. Therefore, we next investigated whether SPHKAP and RI intrinsically form such structures using an array of SPHKAP and RIα constructs (Fig. 4a) heterologously expressed in HEK293T (HEK) cells. We used previously characterized GFP-tagged RIα constructs[27,28], including full-length RIα, as well as variants including or lacking the AKAP-interacting dimerization and docking (D/D) domain. We co-expressed these RIα-GFP constructs with full-length mScarlet-tagged SPHKAP or two deletion mutants, one lacking both RI anchoring domains (SPHKAP-N-mScarlet), or one lacking the phospho-FFAT motif (SPHKAP-C-mScarlet). At low expression levels SPHKAP-mScarlet showed reticular localization reminiscent of ER, while cells with higher SPHKAP-mScarlet expression occasionally contained large intracellular puncta (Fig. 4b); GFP-tagged RIα (RIα-GFP) formed

small puncta in HEK cells (Supplementary Fig. 6a), as previously described[27,28]. Co-expression of SPHKAP-mScarlet and RIα-GFP dramatically increased the size of their respective puncta (Fig. 4b–d). A similar increase in SPHKAP cluster size was observed in HEK cells expressing untagged SPHKAP and HA-tagged RIα (Supplementary Fig. 6a, b), supporting that a synergistic increase in SPHKAP and RIα cluster size is due to the intrinsic properties of these proteins resulting from their co-expression. These results demonstrated that large SPHKAP-RI co-clusters are qualitatively similar to those observed endogenously in brain neurons could be recapitulated in heterologous cells.

We next evaluated the necessity of SPHKAP's and RI's respective interaction domains in driving formation of these large co-clusters. We found that the AKAP- and RI-interacting D/D domain of RIα was by itself sufficient to increase SPHKAP cluster size, while an RIα construct lacking the D/D domain was not (Fig. 4b–d). Similarly, a SPHKAP truncation mutant containing both of its RI-anchoring domains (SPHKAP-C) showed a large increase in cluster size when co-expressed with RIα, while a mutant lacking the RI-interacting domains (SPHKAP-N) was unaffected by RIα co-expression (Fig. 4e and Supplementary Fig. 6c).

Clustering of heterologously expressed RIα has previously been reported to be sensitive to disruption of the AKAP-binding D/D domain of RIα, elimination of the identified responsible AKAP (AKAP11), or interference with the RI-AKAP11 interaction[27]. We confirmed that expression of RI anchoring disruptor (RIAD)[29], a peptide inhibitor of AKAP-RI binding, abolished RIα-GFP clustering and its association with endogenous AKAP11 in HEK cells (Supplementary Fig. 6d, e). Similarly, RIAD prevented reciprocal co-clustering of SPHKAP and RIα, whereas a peptide inhibitor of AKAP-RII interaction (sAKAP-*IS*)[30], the scrambled

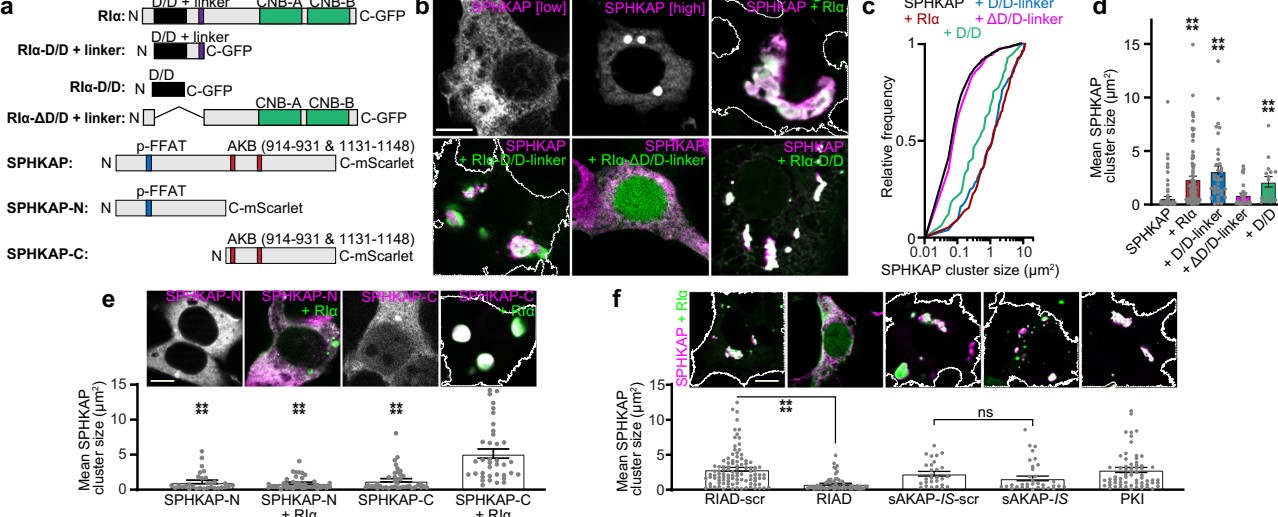

**Fig. 4 | RI anchoring drives SPHKAP-RI co-clustering. a** Schematic of RIα-GFP (D/D dimerization and docking domain [black], pseudosubstrate inhibitor domain [purple], CNB-A and CNB-B cyclic nucleotide binding domains [green]) and SPHKAP-mScarlet constructs (p-FFAT phospho-FFAT motif [blue], AKB A-kinase binding domains [red]). **b** Representative images of HEK293T (HEK) cells quantified in **c-d** expressing full-length SPHKAP-mScarlet (magenta) and/or RIα-GFP and RIα-GFP D/D domain mutants (green). Scale bar, 10 μm. **c** Quantification of pooled SPHKAP-mScarlet mean ± s.e.m. cluster size in all HEK cells co-expressing indicated RIα-GFP constructs. Data are mean ± s.e.m. area of 1045 puncta from 87 cells (SPHKAP), 168 puncta from 97 cells (+RIα), 115 puncta from 35 cells (+D/D-linker), 588 puncta from 37 cells (+RIα-ΔD/D-linker), and 52 puncta from 15 cells (+D/D). Statistical significance determined by one-way ANOVA followed by Dunn's multiple comparisons test (vs. SPHKAP); ****P < 0.0001. **e** Representative images of HEK cells expressing SPHKAP-mScarlet deletion mutants (magenta) and RIα-GFP (green)

(scale bar, 10 μm); mean ± s.e.m. SPHKAP cluster size for each condition is quantified below the representative images for 31 (SPHKAP-N), 41 (SPHKAP-N + RIα), 40 (SPHKAP-C), and 37 (SPHKAP-C + RIα) cells; each point represents 1 cell. Statistical significance determined by one-way ANOVA followed by Dunn's multiple comparisons test (vs. SPHKAP-C + RI); ****P < 0.0001. **f** Representative images of HEK cells expressing full-length SPHKAP-mScarlet (magenta), RIα-GFP (green), and the indicated interfering peptide (not shown) (scale bar, 10 μm); mean ± s.e.m. SPHKAP cluster size for each condition is quantified below the representative images for 100 (RIAD-scr), 65 (RIAD), 28 (sAKAP-*IS*-scr), 32 (sAKAP-*IS*), and 66 (PKI) cells; each point represents 1 cell. Statistical significance determined by one-way ANOVA followed by Dunn's multiple comparisons test (vs. indicated pairs); ****P < 0.0001; ns not significant (P = 0.5189). PKI compared to either scr construct was not significant (P > 0.9999). Source data are provided as a Source Data file.

controls of RIAD and sAKAP-IS (RIAD-scr and sAKAP-IS-scr, respectively) and PKI, a pseudosubstrate inhibitor of PKA-C catalytic activity, were without effect (Fig. 4f). RIAD also reduced endogenous SPHKAP cluster size in neurons (Supplementary Fig. 7a, b), further supporting the role of the SPHKAP-RI interaction in their reciprocal co-clustering in brain neurons.

In addition to AKAP binding[27], RIα-GFP clustering has also been reported to be a consequence of liquid–liquid phase separation (LLPS)[28]. Given these results, as well as the qualitatively "droplet"-like appearance of SPHKAP-RI co-clusters, we tested the hypothesis that the physical properties of SPHKAP-RI puncta would be consistent with LLPS. We treated HEK cells expressing SPHKAP-mScarlet and cultured hippocampal neurons with 1,6-hexanediol, which disrupts certain liquid-like protein assemblies[31]. Hexanediol treatment had little effect on SPHKAP-RI cluster properties in either HEK cells (Supplementary Fig. 7c–e) or neurons (Supplementary Fig. 7f–h). Elevation of cAMP has previously been found to increase clustering of RIα-GFP[27,28], thus we assessed how treatment of neurons with the cAMP-elevating agents forskolin and IBMX affected SPHKAP-RI assemblies. Interestingly, rather than enhancing the clustering of SPHKAP-RI, elevation of cAMP by forskolin/IBMX treatment unexpectedly resulted in significant disassembly of SPHKAP-RI clusters and their spatial uncoupling from VAPs (Supplementary Fig. 7f–i). Forskolin/IBMX-mediated disassembly of SPHKAP complexes still occurred when PKA catalytic activity was blocked (Supplementary Fig. 7f–i), suggesting that supraphysiological elevation of cAMP mediated by these agents[32] rather than PKA-dependent phosphorylation causes SPHKAP-RI complexes to decluster in neurons. Taken together, our data suggest that when RI is associated with an AKAP, such as SPHKAP, it can acquire physical properties distinct from previously described LLPS-derived RI condensates[28].

We also measured fluorescence recovery after photobleaching (FRAP) as a further test of the physical state of SPHKAP-RI complexes. Neither SPHKAP-mScarlet nor RIα-GFP fluorescence recovered after photobleaching a portion of or the entire cluster in neurons (Supplementary Fig. 8a–e) or HEK cells (Supplementary Fig. 8f–k), indicating little mobility of SPHKAP or RI within these assemblies. SPHKAP-mScarlet co-expression with RIα-GFP further limited the mobility of both proteins in neurons (Supplementary Fig. 8b, c) and HEK cells (Supplementary Fig. 8h, i). In contrast, soluble DsRed-Monomer had a uniform distribution throughout the cytoplasm when co-expressed with SPHKAP-tagBFP and RI-HA, including throughout areas containing large SPHKAP-tagBFP clusters (Supplementary Fig. 9a). FRAP analysis of SPHKAP and DsRed revealed that soluble DsRed was mobile throughout the relatively immobile SPHKAP cluster (Supplementary Fig. 9b). Thus, our results show that SPHKAP-RI clusters are stable assemblies permeable to molecules at least as large as DsRed-Monomer (a 2.3 by 4.2 nm cylinder[33]). Taken together, these results imply that signaling molecules such as PKA-C can diffuse in and out of SPHKAP-RI assemblies, suggesting they serve as large and stable reservoirs for type I PKA near ER-PM junctions.

### SPHKAP-RI complexes associate with VAPs

Like Kv2.1 and STARD3, a cholesterol transfer protein, SPHKAP possesses a putative phospho-FFAT motif (Fig. 5a)[22]. Because we observed enrichment of VAP immunolabeling at SPHKAP clusters in brain (Supplementary Fig. 3) and cultured neurons (Supplementary Fig. 7i), and because immuno-EM revealed that endogenous SPHKAP-RI assemblies tightly associate with stacked ER, we examined whether SPHKAP impacted VAP protein organization and ER structure when exogenously expressed. Endogenous VAPs were recruited to exogenous SPHKAP-RI clusters in HEK cells (Fig. 5b) and neurons (Supplementary Fig. 10). SPHKAP mutants lacking the phospho-FFAT motif (SPHKAP-C) or with point mutations at either of two residues (F212, T214) predicted to be critical for phospho-FFAT interaction with VAPs[6,22] eliminated co-localization with VAPs, although all mutant

constructs displayed robust co-clustering with RIα-GFP and endogenous PKA-C (Fig. 5b, c). Introduction of a phosphomimetic aspartate residue at SPHKAP T214 failed to recover the SPHKAP-VAP spatial association, consistent with previous findings that phosphomimetic mutations do not typically meet the structural constraints required to restore the phospho-FFAT-VAP interaction[22]. HEK cells co-expressing SPHKAP-mScarlet and RIα-GFP contained arrays of organized smooth ER (OSER[34], Fig. 5d) absent in untransfected cells or cells co-expressing RIα-GFP and an FFAT-motif-lacking SPHKAP-C-mScarlet truncation mutant.

To test whether endogenous SPHKAP and VAPs physically interact, we designed interfering peptides in the form of two cell-penetrating TAT peptides containing the canonical FFAT motif (SEDEFYDALS) from oxysterol-binding protein-related protein 1 (ORP1), whose binding to VAPA has been structurally characterized[35], and treated cultured hippocampal neurons with these peptides overnight. We found that these TAT-FFAT peptides dissociated SPHKAP from VAPs, but a scrambled control TAT peptide was without effect (Fig. 5e, f). Like FFAT mutant SPHKAP constructs expressed in HEK cells, these TAT-FFAT interfering peptides did not impact SPHKAP clustering, demonstrating that formation of SPHKAP-RI assemblies can be dissociated from their interaction with VAPs. Together, these results support that an intact phospho-FFAT motif in SPHKAP is required for SPHKAP-RI assembly-dependent ER rearrangement and concentration in the cytosolic space between stacked ER (Fig. 3 and Supplementary Fig. 5). A proposed model for the association of SPHKAP-RI assemblies and VAPs with stacked ER cisternae is shown in Fig. 5g.

### SPHKAP supports LTCC-dependent Ca²⁺ entry and transcription factor activation

The activity of PM LTCC and ER RyR $Ca^{2+}$ channels, enriched components of Kv2.1-containing ER-PM junctions[9,11], is enhanced by PKA phosphorylation[36,37]. AKAP-mediated anchoring of PKA near LTCCs and RyRs is an important determinant of the activity of these channels in neurons[38] and cardiomyocytes[39]. Immunolabeling revealed SPHKAP and type I PKA near Cav1.2 channels and RyRs associated with Kv2.1 at ER-PM junctions in hippocampal neurons (Fig. 6a and Supplementary Fig. 11a–c). We used PLA to evaluate whether SPHKAP placed type I PKA near neuronal RyRs, which are spatially and functionally coupled with LTCCs at somatic ER-PM junctions formed by Kv2.1[9]. We observed prominent RyR-RI PLA signal in control neurons, but not in SPHKAP KD neurons or in single primary antibody control neurons (Fig. 6b). While SPHKAP KD did not affect the overall abundance of PKA-C in cultured hippocampal neurons (Supplementary Fig. 11d), SPHKAP KD decreased the number of PKA-C subunits in close proximity to Cav1.2 (Supplementary Fig. 11e, f).

We next examined how reduced SPHKAP-dependent anchoring of type I PKA near RyRs and Cav1.2 impacted depolarization-dependent somatic $Ca^{2+}$ entry. Using a depolarization paradigm that maximizes the contribution of voltage-gated $Ca^{2+}$ channels to overall depolarization-triggered $Ca^{2+}$ entry[40] we found that $Ca^{2+}$ influx induced by depolarization with elevated extracellular $K^+$ was impaired in SPHKAP KD neurons compared to scrambled (scr) shRNA-transfected controls (Fig. 6c, e and Supplementary Fig. 11g, h). These results are consistent with the observation that direct inhibition of PKA or its uncoupling from AKAPs suppresses LTCC activity[41]. In contrast, $Ca^{2+}$ influx elicited in the presence of the LTCC blocker nimodipine was similar between control and SPHKAP KD neurons (Fig. 6d, f). These results indicate that SPHKAP supports nimodipine-sensitive (i.e., LTCC-mediated) $Ca^{2+}$ influx in hippocampal neurons.

LTCC-dependent $Ca^{2+}$ signals originating from Kv2.1-associated ER-PM contacts lead to phosphorylation of cAMP response element-binding protein (CREB) and expression of the immediate early gene c-Fos[11], transcription factors with key roles in learning and memory. In

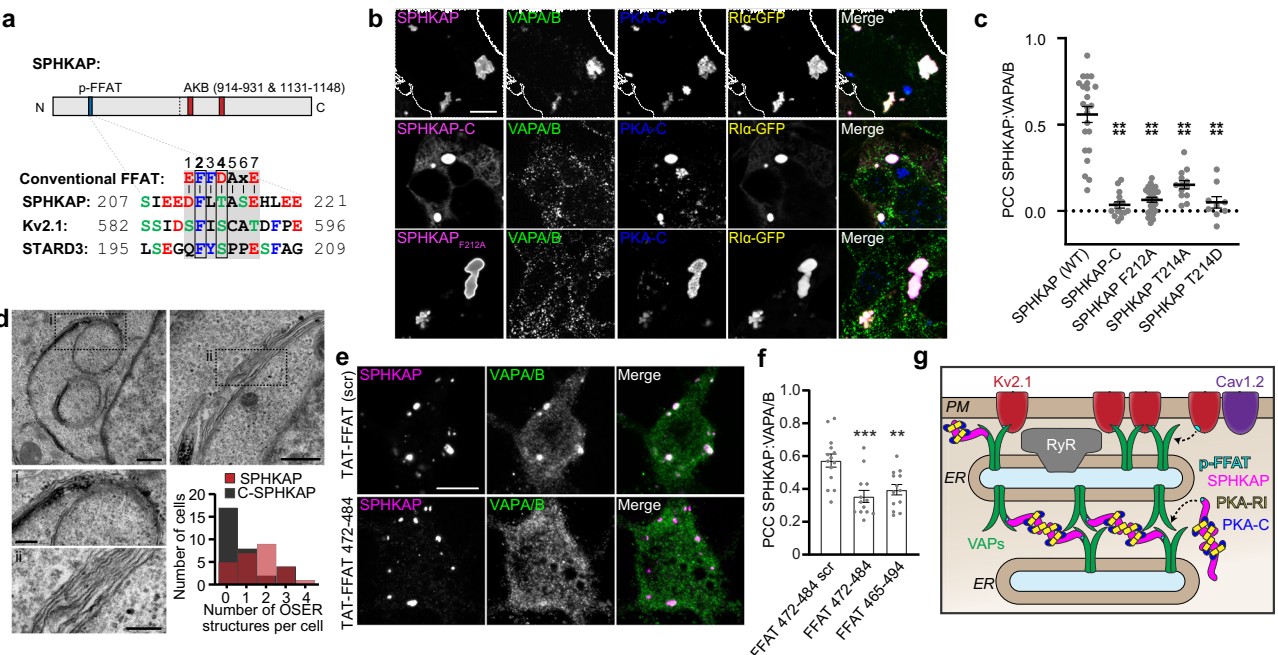

**Fig. 5 | SPHKAP-RI complexes associate with VAPs. a** Alignment of a conventional FFAT motif with phospho-FFAT motifs (7 core residues highlighted) found in SPHKAP, Kv2.1, and STARD3. Alcoholic residues are highlighted in green, acidic residues in red, and aromatic residues in blue; all other residues are in black. Critical F residue at position 2 (corresponding to SPHKAP F212) and T residue at position 4 (corresponding to SPHKAP T214) in these FFAT motifs are outlined with boxes. **b** Representative images of HEK cells (quantified in **c**) expressing WT SPHKAP-mScarlet, SPHKAP-C-mScarlet, or SPHKAP-mScarlet F212A, RIα-GFP, and immuno-labeled for endogenous VAPs and PKA-C. Scale bar, 10 μm. **c** Quantification of SPHKAP-mScarlet overlap with endogenous VAPs (mean ± s.e.m.; Pearson's corre-lation coefficient, PCC) in HEK cells transfected with the indicated SPHKAP-mScarlet constructs; each point represents 1 cell (n = 23 [WT], 15 [SPHKAP-C], 25 [F212A], 13 [T214A], and 9 [T214D] cells). Statistical significance determined by one-way ANOVA followed by Tukey's multiple comparisons test (vs. WT); ****P < 0.0001. **d** Representative TEM images of OSER in HEK cells co-transfected with SPHKAP-

mScarlet and RIα-GFP and histogram of number of OSER structures/cell in single EM sections acquired from HEK cells transfected with SPHKAP-mScarlet + RIα-GFP (n = 26 cells) or SPHKAP-C-mScarlet + RIα-GFP (n = 30 cells). Scale bars, 500 nm. **e** Representative images of neurons treated with control TAT-FFAT-scr or TAT-FFAT 472–484 peptides overnight and then immunolabeled for endogenous SPHKAP and VAPs (quantified in **f**). Scale bar, 10 μm. **f** Colocalization (Pearson's colocalization coefficient, PCC) of VAPs with SPHKAP under indicated conditions. Each point represents 1 cell (n = 14 [scr], 13 [472–484], and 13 [465–494] cells). Statistical significance determined by one-way ANOVA followed Tukey's multiple comparisons test (vs. scr); ***P = 0.0004, **P = 0.0038. Source data are provided as a Source Data file. **g** Schematic illustrating hypothesized mechanism of SPHKAP-RI organization at Kv2.1-associated ER-PM junctional domains: phosphorylation of the phospho-FFAT (p-FFAT) motif in Kv2.1 or SPHKAP leads to their physical associa-tion with VAPs, while RI association with SPHKAP enables formation of SPHKAP-RI oligomers.

---

addition, shRNA-mediated knockdown of PKA-RIβ, but not -RIIβ, was previously found to impair forskolin-induced CREB phosphorylation in cultured hippocampal neurons[14]. We found that SPHKAP KD impaired depolarization-induced CREB and c-Fos activation (Fig. 7a–c), sug-gesting a key role for SPHKAP anchored type I PKA in supporting excitation-transcription coupling. Together, these results show that SPHKAP localizes type I PKA near Cav1.2 and RyR channels at Kv2.1-organized ER-PM junctions to create a somatic PKA-regulated signaling complex that promotes depolarization-induced $Ca^{2+}$ signaling and transcription factor activation.

### SPHKAP underlies PKA-dependent excitation-phosphorylation coupling

$Ca^{2+}$ can activate PKA in hippocampal neurons through cAMP gen-erated by $Ca^{2+}$/calmodulin-activated adenylyl cyclases[42,43], increas-ing somatic voltage-gated $Ca^{2+}$ channel open probability[44]. $Ca^{2+}$ is also hypothesized to enhance type I PKA activity by impairing binding of RI's pseudosubstrate inhibitor domain to PKA-C[45]. To investigate whether type I PKA localization near LTCCs impacts its function, we first quantified depolarization-induced LTCC-depen-dent activation of somatic PKA in neurons (Fig. 8a). We found that increasing depolarization via progressively elevated external $K^+$ correlated with greater PKA substrate phosphorylation that was blocked by inhibition of LTCCs (nimodipine) or PKA catalytic activity (H89) (Fig. 8b).

LTCC-dependent activation of PKA also decreased PKA-C asso-ciation with SPHKAP and reduced PKA-C cluster size (Fig. 8c). These changes were prevented by blocking LTCCs with nimodipine and mimicked by generating cAMP via activation of β-adrenergic receptors with isoproterenol. Depolarization-dependent changes in PKA-C clus-ter size were not observed in GABAergic interneurons, identified by GAD67 immunolabeling, which lacked SPHKAP immunolabeling and displayed much smaller PKA-C clusters than GAD67-negative neurons under basal conditions (Fig. 8c). While depolarization diminished PKA-C clustering, neither SPHKAP cluster size nor distribution were sub-stantially affected (Supplementary Fig. 12a). These results further support that liberated PKA-C molecules are mobile within SPHKAP-RI assemblies, presumably enabling them to phosphorylate nearby sub-strates. We previously showed that uncoupling of Cav1.2 from Kv2.1-mediated ER-PM junctions using the cell-permeant peptide TAT-CCAD impaired $Ca^{2+}$ signaling at these sites[11]. TAT-CCAD treatment also blocked depolarization-induced dissociation of PKA-C from SPHKAP (Supplementary Fig. 12b), indicating that LTCC-mediated $Ca^{2+}$ entry at Kv2.1 ER-PM junctions activates PKA.

We next tested whether decoupling type I PKA from SPHKAP with the RIAD interfering peptide impacted its LTCC-dependent activation. We used ExRai-AKAR2[46] to image PKA activity in neurons expressing either RIAD, RIAD-scr, or the PKA catalytic activity inhibitor PKI. LTCC-dependent PKA activity was suppressed in neurons expressing RIAD or PKI (Fig. 8d). However, RIAD had no effect on PKA activity induced by

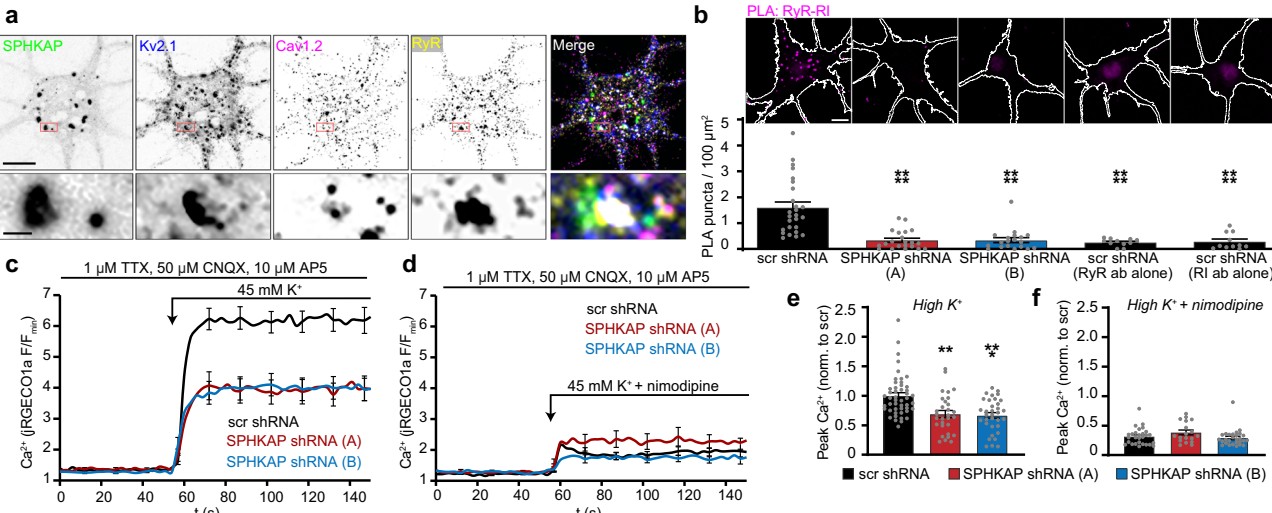

**Fig. 6 | SPHKAP supports LTCC-dependent Ca²⁺ entry. a** Images of a hippocampal neuron (representative of *n* = 6 neurons) immunolabeled for SPHKAP (green), Kv2.1 (blue), Cav1.2 (magenta), and RyR (yellow). Scale bar, 10 μm. **b** Representative images of proximity ligation assay (PLA) puncta (magenta) detected in control neurons expressing a scrambled shRNA (scr) or in neurons expressing one of two distinct anti-SPHKAP shRNAs (A or B) and immunolabled for RyRs and RI. The neuron periphery is outlined in each image; scale bar, 10 μm. Quantification of mean ± s.e.m PLA signal density for each condition is presented below each image. Each point represents 1 cell; *n* = 26 (scr), 21 (A), 19 (B), 11 (RyR ab alone), and 11 (RI ab alone) cells; statistical significance was determined using one-way ANOVA followed by Tukey's multiple comparisons test (vs. scr); ****$P$ < 0.0001. **c** Mean ± s.e.m. jRGECO1a fluorescence in neurons expressing either control scr shRNA (black, *n* = 48 cells) or anti-SPHKAP shRNAs (shRNA A, red, *n* = 30 cells; shRNA B, blue,

*n* = 34 cells) and depolarized with 45 mM external K⁺. **d** Mean ± s.e.m. jRGECO1a fluorescence in neurons expressing either control scr shRNA (black, *n* = 28 cells) or anti-SPHKAP shRNAs (shRNA A, red, *n* = 18 cells; shRNA B, blue, *n* = 28 cells) and depolarized with 45 mM external K⁺ and 10 μM nimodipine. **e** Mean ± s.e.m peak jRGECO1a fluorescence (normalized to response in scr cells) in cells depolarized with 45 mM external K⁺; each point represents 1 cell (*n* cells as in **c**); statistical significance was determined using one-way ANOVA followed by Dunn's multiple comparisons test (vs. scr); ***$P$ = 0.0002, **$P$ = 0.0004. **f** Mean ± s.e.m peak jRGECO1a fluorescence (normalized to response to 45 mM K⁺ in scr cells) in cells depolarized with 45 mM external K⁺ in the presence of 10 μM nimodipine; each point represents 1 cell (*n* cells as in **d**); statistical significance was assessed using one-way ANOVA followed by Dunn's multiple comparisons test (vs. scr); $P$ = 0.9192 (A), $P$ > 0.9999 (B).

treatment with the cAMP-elevating drugs forskolin and IBMX (Fig. 8e), suggesting that LTCC-mediated Ca²⁺ influx preferentially activates SPHKAP-anchored type I PKA. Consistent with this, depolarization- but not forskolin/IBMX-induced elevations in PKA substrate phosphorylation were impaired in SPHKAP KD neurons (Supplementary Fig. 12c, d). Finally, we found that depolarization-dependent elevations in ExRai-AKAR2 fluorescence were absent in SPHKAP KD neurons or in the presence of the LTCC blocker nimodipine (Fig. 8f and Supplementary Fig. 12e). Together, our results define SPHKAP-containing neuronal ER-PM junctional domains as specific sites underlying reciprocal coupling of LTCC-dependent Ca²⁺ and PKA signaling in neuronal somata and for linking membrane depolarization to PKA phosphorylation of downstream targets to support excitation-phosphorylation coupling (Fig. 8g).

## Discussion

By investigating molecules associated with Kv2.1 in brain neurons we discovered a PKA signalosome associated with somatic ER-PM junctions. A central component of this signalosome is the type I PKA-specific anchoring protein SPHKAP, which tethers type I PKA to ER cisternae at ER-PM junctions. AKAPs direct PKA to defined subcellular locations[12], and our results support that assembly of the SPHKAP-type I PKA signaling system on the ER near the Ca²⁺ signaling machinery at Kv2.1-associated ER-PM junctions enables depolarization-regulated PKA signaling distinct from that of neurotransmitter receptor-regulated type II PKA anchored in dendrites[13]. Moreover, SPHKAP's structural capacity to form highly concentrated type I PKA assemblies while simultaneously mediating interactions with the ER scaffolding protein VAP allows PKA enrichment on the ER between stacked subsurface cisternae. Altogether, these results establish ER-PM junctional domains as "signaling islands[12]" that organize and regulate PKA signaling in the

neuronal soma and provide insight into mechanisms controlling the subcellular localization and regulation of type I PKA.

Our finding of a Ca²⁺-activated PKA signaling apparatus assembled near Kv2.1-organized ER-PM junctions supports the emerging picture that these domains integrate diverse signaling pathways triggered by a common stimulus−neuronal firing. Somatic subsurface cisternae have narrow (10−20 nm) gaps between the PM, ER, and subsequent stacks of ER cisternae[4] that enables significant concentration of signaling molecules in the cytosolic space between membranes, enhancing the kinetics and amplitude of their responses. The formation of SHPKAP-type I PKA assemblies and their simultaneous enrichment between ER cisternae profoundly increases the local type I PKA concentration at these sites. As previously suggested[16], this enrichment may be essential for sensitizing second messenger responses within the SPHKAP-type I PKA nanodomain. The assembly of these molecules at sites where PM ion channels are clustered to generate compartmentalized Ca²⁺ signals, combined with the lower cAMP concentrations required to activate type I PKA as compared to type II PKA[47], supports that SPHKAP-type I PKA is poised to couple membrane depolarizations with PKA activation. Neurons that express SPHKAP-type I PKA signaling complexes may have specific physiological requirements for rapid coupling of firing to PKA activity. This may be important in neuronal somata that integrate the activity of thousands of synaptic inputs into firing outputs and changes in gene expression. Indeed, brain neurons expressing a dominant-negative form of PKA-RI that specifically inhibited type I PKA activity displayed impaired CREB phosphorylation and c-Fos expression[48]. In addition, genetic disruption of RIβ was found to impair synaptic plasticity in hippocampal CA1 pyramidal neurons and dentate gyrus granule cells[49], both of which display prominent SPHKAP-type I PKA expression.

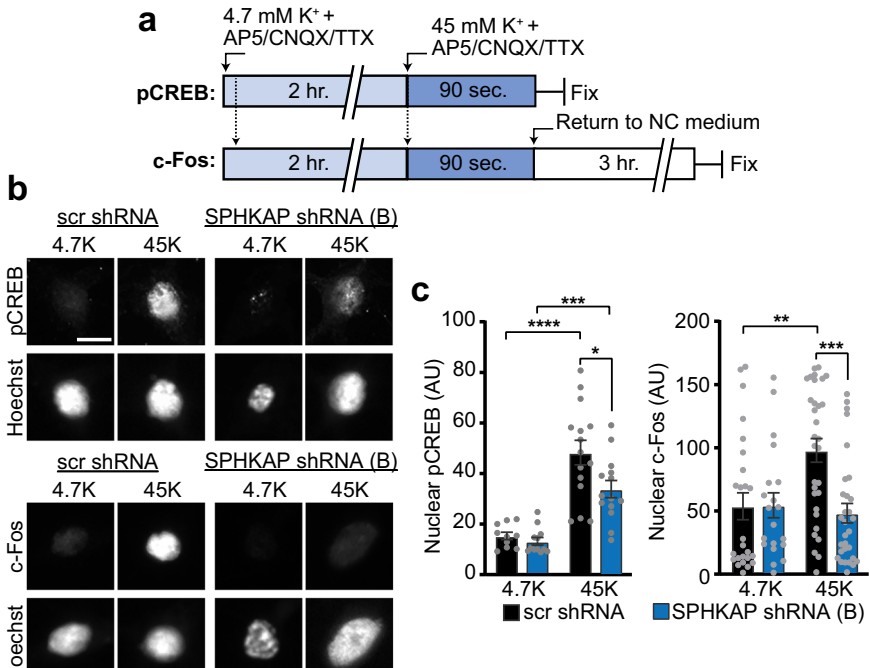

**Fig. 7 | SPHKAP promotes depolarization-induced transcription factor activation. a** Experimental paradigm to assess depolarization-induced transcription factor activation in cultured hippocampal neurons. **b** Images of phosphorylated CREB (pCREB) and c-Fos immunolabeling in control (4.7 mM $K^+$ solution: 4.7 K) and depolarized (45 mM $K^+$ solution: 45 K) neurons expressing scr or SPHKAP shRNA B. Scale bar, 10 μm. **c** Mean ± s.e.m pCREB and c-Fos immunofluorescence signal intensity; each point represents 1 cell ($n = 10$ [scr, 4.7 K pCREB], 12 [SPHKAP, 4.7 K pCREB], 15 [scr, 45 K pCREB], 14 [SPHKAP, 45 K pCREB], 25 [scr, 4.7 K c-Fos], 31 [SPHKAP, 4.7 K c-Fos], 20 [scr, 45 K c-Fos], and 30 [SPHKAP, 45 K c-Fos] cells. Statistical significance determined using one-way ANOVA followed by Tukey's multiple comparisons test; *$P = 0.0185$, **$P = 0.0051$, ***$P = 0.008$ (pCREB) or 0.006 (c-Fos), ****$P < 0.0001$. Source data are provided as a Source Data file.

Our results also demonstrate that in addition to its established signaling role, type I PKA can function as a structural protein when assembled with SPHKAP. Formation of large (micron-sized) SPHKAP-RI assemblies requires anchoring of RI to SPHKAP, as interference with either protein's respective interaction domain prevented the formation of SPHKAP-RI clusters. SPHKAP is among a small subset of AKAPs that can accommodate two PKA holoenzymes[16], and we suggest that this property enables formation of large SPHKAP-RI assemblies. Interestingly, AKAP11, a paralog of SPHKAP, can also bind two type I PKA holoenzymes[50], and interfering with type I PKA anchoring also prevents formation of endogenous clustered AKAP11 assemblies formed by heterologous expression of RI. Unlike AKAP11-RI assemblies, whose clustering is augmented by elevation of cAMP[27], we find that neuronal SPHKAP-RI assemblies are declustered by cAMP. This may serve to dampen PKA signaling under conditions of excessive neuronal activity, paralleling the disassembly of Kv2.1[51] and Cav1.2[9] channels from ER-PM junctions when neurons are strongly depolarized and further underscoring the robust activity-dependent structural plasticity of these domains.

Like Kv2.1, SPHKAP also possesses a phospho-FFAT motif, enabling phosphorylation-dependent reversible association of SPHKAP-type I PKA assemblies with the ER. Our data suggest that the VAP-binding property of SPHKAP allows SPHKAP-type I PKA assemblies to generate stacked ER, organizing cellular structure and concentrating type I PKA on the ER. These results also reinforce the central role of VAPs as versatile scaffolds mediating formation of ER-organelle membrane contacts through interactions with diverse FFAT-motif containing proteins. Determining whether other PKA-AKAP or AKAP-VAP complexes serve similar dual structural and signaling roles may reveal a modular mechanism for interactions associating organelle structures with signaling pathways that is likely to have broad physiological significance.

## Methods

### Animals

All procedures using mice and rats were approved by the University of California, Davis Institutional Animal Care and Use Committee and performed in accordance with the NIH Guide for the Care and Use of Laboratory Animals. Animals were maintained under standard light-dark cycles and allowed to feed and drink ad libitum. Experiments using hippocampal neurons were performed using cultures obtained from pooling neurons from Sprague-Dawley rat (Charles River Strain Code: 001) fetuses of both sexes. Adult C57BL/6 J mice 8–12 weeks old of both sexes were used in immunohistochemistry and proteomic experiments.

### Antibodies

All primary antibodies used in this study are listed in Supplementary Table 1. Anti-SPHKAP rabbit polyclonal (pAb) and mouse monoclonal antibodies (mAbs) were generated for this study. In brief, rabbits and mice were immunized with a recombinant fragment corresponding to the C-terminal 126 amino acids (a.a. 1533–1658) of mouse SPHKAP (Uniprot accession E9PUC4). This sequence is present in all SPHKAP splice variants except the C-terminally truncated isoform Q6NSW3-4. Rabbit antisera was generated at Pocono Rabbit Farm (Canadensis, PA). Anti-SPHKAP pAbs were affinity purified from the antisera by strip purification against nitrocellulose membranes onto which 2 mgs of this recombinant SPHKAP protein had been electrophoretically transferred from an SDS curtain gel[52]. Anti-SPHKAP mAbs were produced from hybridomas generated from splenocytes obtained from two Balb/c mice immunized with this recombinant SPHKAP protein using standard methods[53,54]. The anti-SPHKAP mouse mAbs were selected following evaluation by ELISA, immunocytochemistry against transfected COS-1 cells exogenously expressing full-length SPHKAP protein and immunoblots and immunohistochemistry against rat brain samples[55]. Specificity was validated by shRNA-mediated knockdown of

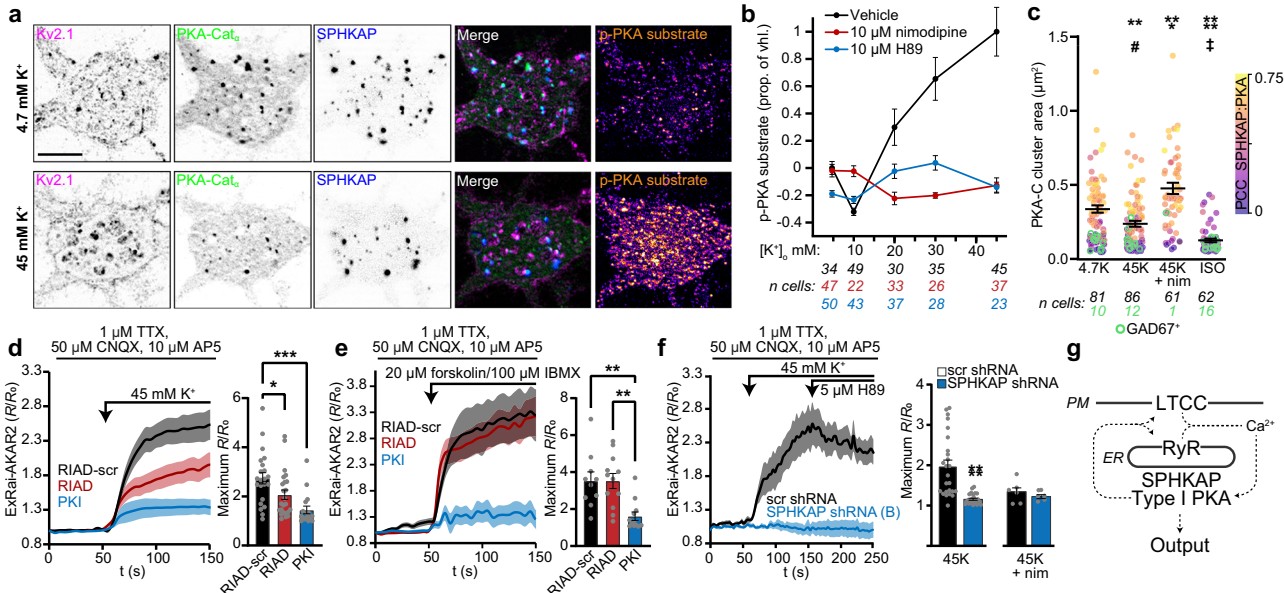

**Fig. 8 | SPHKAP underlies PKA-dependent excitation-phosphorylation coupling. a** Representative images of control (4.7 mM K$^+$) and depolarized (45 mM K$^+$) neurons (quantified in **b**) immunolabeled for Kv2.1 (magenta), PKA-C (green), SPHKAP (blue), and phosphorylated PKA (p-PKA) substrate (orange). Scale bar, 10 μm. **b** Mean ± s.e.m. p-PKA substrate immunofluorescence signal in neurons stimulated with elevated extracellular potassium ([K$^+$]$_o$); $n$ cells analyzed under each condition noted below graph (black: vehicle; red: nimodipine; blue: H89). **c** PKA-C cluster size (points) and spatial association with SPHKAP (Pearson's correlation coefficient [PCC], color gradient) in neurons stimulated with 4.7 mM K$^+$ (4.7 K), 45 mM K$^+$ (45 K), 45 mM K$^+$ + 10 μM nimodipine (45 K + nim), or 1 μM isoproterenol (ISO). GAD67$^+$ cells: green symbols. Data are mean ± s.e.m. (excluding GAD67$^-$ cells); each point represents 1 cell; $n$ cells analyzed under each condition is noted below the graph (black: GAD67$^-$ cells, green: GAD67$^+$ cells). Statistical significance of differences in PKA-C cluster size or PCC in GAD67$^-$ cells assessed using

one-way ANOVA followed by Dunnett's multiple comparisons test (vs. 4.7 K); PKA-C cluster size: **$P = 0.0072$, ***$P = 0.0004$, ****$P < 0.0001$; PCC: $^\#P = 0.0334$; $^\ddagger P < 0.0001$. **d** Left, mean ± s.e.m. ExRai-AKAR2 fluorescence in neurons expressing PKI, RIAD, or RIAD-scr. Right, peak fluorescence from data in left panel. Each point represents 1 cell ($n = 22$ [RIAD-scr], 24 [RIAD], and 17 [PKI] cells). Statistical significance determined using one-way ANOVA followed by Tukey's multiple comparisons test; *$P = 0.0443$, **$P = 0.0051$, ***$P = 0.002$. **e** As in **d**, in neurons stimulated with forskolin and IBMX ($n = 10$ [RIAD-scr], 12 [RIAD], and 12 [PKI] cells); **$P = 0.0034$ (RIAD-scr versus PKI) and **$P = 0.0023$ (RIAD versus PKI). **f** as in **d**, in neurons expressing scr or SPHKAP shRNA ($n = 25$ [scr, 45 K], 25 [SPHKAP, 45 K], 7 [scr, 45 K + nim], and 5 [SPHKAP, 45 K + nim] cells). Statistical significance determined using unpaired two-tailed Student's $t$-test; ****$P < 0.0001$. **g** Hypothesized mechanism of depolarization-induced activation of PKA and regulation of Ca$^{2+}$ signaling machinery. Source data are provided as a Source Data file.

---

endogenous SPHKAP expressed in cultured hippocampal neurons (Supplementary Fig. 4). All antibodies not available through other sources are available from the corresponding authors.

## DNA constructs
SPHKAP-mScarlet (WT, F212A, T214A, and T214D) and SPHKAP-TagBFP plasmids were generated by GenScript, cloning mouse SPHKAP (accession number NM_172430.3) from pCMV6-SPHKAP-Myc-DDK (OriGene Cat# MR218656) and mScarlet[56] or TagBFP[57] into the pEGFP-N1 backbone (Clontech) from which the DNA encoding eGFP was excised. pCMV6-SPHKAP-Myc-DDK contained a spurious nucleotide sequence encoding 49 amino acids not corresponding to the SPHKAP accession sequence; this and the sequence encoding the Myc-DDK tag were removed to generate untagged SPHKAP, SPHKAP-mScarlet, and SPHKAP-tagBFP constructs. SPHKAP-N-mScarlet and SPHKAP-C-mScarlet were generated by cloning DNA encoding the first 819 amino acids (SPHKAP-N) or amino acids 820–1658 (SPHKAP-C) of SPHKAP and mScarlet into the pEGFP-N1 backbone. Plasmids encoding anti-rat-SPHKAP and scrambled control shRNAs were obtained from Origene (OriGene Cat# TL708390). Plasmids encoding RIα-GFP (Addgene plasmid# 181840), RIα-D/D + linker-GFP (Addgene plasmid# 181842), RIα-ΔD/D + linker-GFP (Addgene plasmid # 181843), ExRai-AKAR2 (Addgene plasmid# 161753), and RIα-D/D-GFP were gifts of Dr. Jin Zhang. RI-HA was generated by cloning an HA tag onto the 3′ end of human PKA-RIα in the pcDNA3.1(+) vector by GenScript. jRGE-CO1a (Addgene plasmid# 61563) was a gift of Dr. Douglas Kim. VSVG-GFP (Addgene plasmid# 11912) was a gift of Dr. Jennifer Lippincott-Schwartz. RIAD-mCherry, RIAD-scr-mCherry, sAKAP-IS-mCherry,

sAKAP-IS-scr-mCherry, and PKI-mCherry were kindly provided by Dr. Alan Howe.

## Cell culture
**HEK293T cells**. HEK293T cells (ATCC Cat# CRL-3216) were maintained in Dulbecco's modified Eagle's medium (DMEM, Gibco Cat# 11995065) supplemented with 10% Fetal Clone III (HyClone Cat# SH30109.03), 1% penicillin/streptomycin, and 1× GlutaMAX (Thermo-Fisher Cat# 35050061) in a humidified incubator at 37 °C and 5% CO$_2$. Prior to transfection, cells were seeded to poly-L-lysine (Sigma Cat# P1524) -coated microscope cover glasses (VWR Cat# 48366-227) or 96-well glass bottom plates (Cellvis Cat# P96-1.5H-N). Approximately 15 h later, cells were transiently transfected in unsupplemented DMEM using Lipofectamine 2000 (Invitrogen Cat #11668019) following the manufacturer's protocol, then returned to regular growth medium 4 h after transfection. Cells were then used for experiments ~48 h later.

**Hippocampal neurons**. Neuronal cultures were prepared and maintained following the protocol of Kaech and Banker[58]. Hippocampi were dissected from rat embryonic day 18.5 fetuses and enzymatically dissociated for 30 min at 37 °C in HBSS supplemented with 0.25% (w/v) trypsin (Gibco Cat# 15050065), followed by mechanical dissociation via trituration with fire-polished glass Pasteur pipettes. Dissociated cells were plated at a density of 30,000 cells per coverslip onto microscope cover glasses (Karl Hecht Assistent Cat# 92099005050) coated with poly-L-lysine (Sigma Cat# P2636), transferred to wells containing glial support cultures, and maintained in Neurobasal medium (ThermoFisher Cat# 21103049) supplemented with 1x

B27 supplement (Invitrogen Cat# 17504044), 2% GlutaMAX (Thermo-Fisher Cat# 35050061), and 0.001% gentamycin (Gibco Cat# 15710064) and in a humidified incubator at 37 °C and 5% CO$_2$. After 4−5 days in vitro (DIV), cytosine-D-arabinofuranoside (Millipore Cat# 251010) was added to inhibit non-neuronal cell growth. Neurons were transiently transfected at DIV 7−10 using Lipofectamine 2000 for 1 h, then used for experiments 40−48 h post transfection.

## Proteomics

DSP-crosslinked mouse brain samples for immunopurification were prepared as previously described[9] using brains from three pairs of Kv2.1 KO and WT mouse littermates for comparison. Excised brains were homogenized over ice in a Dounce homogenizer containing 5 mL homogenization and crosslinking buffer (in mM): 320 sucrose, 5 NaPO$_4$, pH 7.4, supplemented with 100 NaF, 1 PMSF, protease inhibitors, and 1 DSP (Lomant's reagent, ThermoFisher Cat# 22585). Following a 1-h incubation on ice, DSP was quenched with 20 mM Tris, pH 7.4 (JT Baker Cat# 4109-01 [Tris base]; and 4103-01 [Tris-HCl]). 2 mL of this homogenate was then added to an equal volume of ice-cold 2× radioimmunoprecipitation assay (RIPA) buffer (final concentrations): 1% (vol/vol) TX-100, 0.5% (wt/vol) deoxycholate, 0.1% (wt/vol) SDS, 150 NaCl, 50 Tris, pH 8.0 and incubated for 30 min at 4 °C on a tube rotator. Insoluble material was then pelleted by centrifugation at 12,000×$g$ for 10 min at 4 °C. The supernatant was incubated overnight at 4 °C with the anti-Kv2.1 rabbit polyclonal antibody KC[59] or the anti-Kv1.2 rabbit polyclonal antibody Kv1.2C[60]. Following this incubation, we added 100 μL of magnetic protein G beads (ThermoFisher Cat# 10004D) and incubated the samples for 1 h at 4 °C on a tube rotator. Beads were then washed 6 times following capture on a magnet in ice-cold 1× RIPA buffer, followed by four washes in 50 mM ammonium bicarbonate (Sigma Cat# A6141). Proteins captured on magnetic beads were digested with 1.5 mg/mL trypsin (Promega Cat# V5111) in 50 mM ammonium bicarbonate overnight at 37 °C. The eluate was then lyophilized and resuspended in 0.1% trifluoroacetic acid in 60% acetonitrile.

Proteomic profiling was performed at the University of California, Davis Proteomics Facility. Tryptic peptide fragments were analyzed by LC-MS/MS on a Thermo Scientific Q Exactive Plus Orbitrap Mass spectrometer in conjunction with a Proxeon Easy-nLC II HPLC (Thermo Scientific) and Proxeon nanospray source. Digested peptides were loaded onto a 100 μm × 25 mm Magic C18 100 Å 5U reverse phase trap where they were desalted online, then separated on a 75 μm × 150 mm Magic C18 200 Å 3U reverse phase column. Peptides were eluted using a 60-min gradient at a flow rate of 300 nL per minute. An MS survey scan was obtained for the $m/z$ range 350−1600, where the top 15 ions in the MS spectrum were subjected to HCD (High Energy Collisional Dissociation). Precursor ion selection was performed using a mass window of 1.6 $m/z$, using a normalized collision energy of 27% for fragmentation. A 15 s duration was used for the dynamic exclusion. Tandem MS spectra were extracted and charge state deconvoluted by Proteome Discoverer (Thermo Scientific). Tandem MS samples were then analyzed using X! Tandem (The GPM, thegpm.org; version Alanine (2017. 2. 1.4)). X! Tandem compared acquired spectra against the UniProt Mouse database (May 2017, 103089 entries), the cRAP database of common proteomic contaminants (www.thegpm.org/crap; 114 entries), the ADAR2 catalytic domain sequence, plus an equal number of reverse protein sequences assuming the digestion enzyme trypsin. X! Tandem was searched with a fragment ion mass tolerance of 20 ppm and a parent ion tolerance of 20 ppm. Variable modifications specified in X! Tandem included deamidation of asparagine and glutamine, oxidation of methionine and tryptophan, sulfone of methionine, tryptophan oxidation to formylkynurenin of tryptophan and acetylation of the N-terminus. Scaffold (version Scaffold_4.8.4, Proteome Software Inc., Portland, OR) was used to validate tandem MS-based peptide and protein identifications. X! Tandem identifications were accepted if they possessed -log (expect scores) scores >2.0 with a mass accuracy of 5 ppm. Protein identifications were accepted if they contained at least two identified peptides. The threshold for peptide acceptance was >95% probability.

## Brain fluorescence immunohistochemistry

Following deep anesthesia induced by administration of pentobarbital (Sigma-Aldrich Cat# P3761), animals were transcardially perfused with 4% formaldehyde (freshly prepared from paraformaldehyde, Sigma-Aldrich Cat# 158127) in 0.1 M sodium phosphate buffer pH 7.4 (0.1 M PB). In all, 30 μm-thick sagittal brain sections were prepared and immunolabeled using free-floating methods as detailed previously[19,20]. Sections were permeabilized and blocked in 0.1 M PB containing 10% goat serum and 0.3% Triton X-100 (vehicle) for 1 h at RT, then incubated overnight at 4 °C in primary antibodies diluted in vehicle. After four 5-min washes in 0.1 M PB, sections were incubated with mouse IgG subclass- and/or species-specific Alexa-conjugated fluorescent secondary antibodies and Hoechst 33258 DNA stain (Invitrogen Cat# H2149) diluted in vehicle at room temperature (RT) for 1 h. After two 5-min washes in 0.1 M PB followed by one 5-min wash in 0.05 M PB, sections were mounted and air dried onto gelatin-coated microscope slides, treated with 0.05% Sudan Black (EM Sciences) in 70% ethanol for 2 min[20,61]. Samples were then washed in water and mounted with Prolong Gold (ThermoFisher Cat# P36930). Images of brain sections were taken using the same exposure time to compare the signal intensity directly using a Hamamatsu C11440 sCMOS camera installed on an AxioObserver Z1 microscope with a 10×/0.5 NA lens, Zeiss Collibri 7 LED light source, and an ApoTome2 coupled to Zen software (Zeiss, Oberkochen, Germany). High magnification confocal images were acquired using a Zeiss LSM880 confocal laser scanning microscope equipped with an Airyscan detection unit and a Plan-Apochromat 63×/1.40 NA Oil DIC M27 objective.

## Immunolabeling and imaging of fixed cells

Immunolabeling of cultured hippocampal neurons and HEK293T cells was performed as previously described[6]. Neurons were fixed in ice cold 4% (wt/vol) formaldehyde (freshly prepared from paraformaldehyde, Fisher Cat# O4042) in phosphate-buffered saline (PBS) containing 4% (wt/vol) sucrose (Sigma Cat# S9378), pH 7.4, for 15 min at 4 °C. HEK293T cells were fixed in 3.2% formaldehyde (freshly prepared from paraformaldehyde) prepared in PBS pH 7.4, for 20 min at RT, washed three times for 5 min in PBS. All subsequent steps were performed at RT. Cells were washed three times for 5 min in PBS, followed by blocking in blotto-T (Tris-buffered saline [10 mM Tris, 150 mM NaCl, pH 7.4] supplemented with 4% (w/v) non-fat milk powder and 0.1 % (v/v) Triton-X100 [Roche Cat# 10789704001]) for 1 h. Cells were immunolabeled for 1 h with primary antibodies diluted in blotto-T (concentrations used for primary antibodies listed in Supplementary Table 1). After three 5-min washes in blotto-T, cells were incubated with mouse IgG subclass- and/or species-specific Alexa-conjugated fluorescent secondary antibodies (Invitrogen) diluted in blotto-T for 45 min, then washed 3 × 5 min in PBS. Cover glasses were mounted on microscope slides with Prolong Gold mounting medium. Widefield fluorescence images were acquired on the AxioObserver Z1 microscope described above with a 40×/1.3 NA or 63×/1.40 NA Plan-Apochromat oil immersion objective.

**Super-resolution microscopy.** Neurons were fixed and immunolabeled with primary antibodies as described above. After three 5-min washes in PBS, neurons were incubated for 1 h with Alexa Fluor 647-conjugated goat anti-rabbit IgG or IgG subclass-specific anti-mouse secondary antibody (Invitrogen, 1:1500) and CF568-conjugated goat anti-mouse IgG1 secondary antibody (Biotium Cat# 20248, 1:1500). Cells were then subjected to three 5-min washes in PBS, then mounted on glass depression slides in a GLOX-MEA (Tris 10 mM pH 8, glucose

oxidase 56 mg/mL, catalase 34 mg/mL, 10 mM MEA) solution. Images were acquired using a Leica Infinity TIRF microscope equipped with a ×163/1.49 NA oil immersion objective and a Hamamatsu ORCA flash 4.0 camera using Leica LASX software for image acquisition. Alexa Fluor-647 and CF568 images with 40,000–50,000 cycles and an exposure time of 10 ms per channel were collected with the 638 nm and 561 nm excitation lines, respectively.

**Proximity ligation assay.** The Duolink In Situ PLA kit (Sigma-Aldrich Cat# DUO92008) was used to detect PLA signal between antibody pairs in fixed DIV12-15 transfected with control scrambled or anti-SPHKAP shRNA plasmids. Following fixation as described above, cells were washed in PBS (2 × 3 min) and permeabilized in 0.1% Triton-X100 for 20 min. After blocking at 37 °C for 1 h using Duolink Blocking Solution, cells were incubated overnight at 4 °C with primary antibodies. All other steps were according to manufacturer's instructions. Secondary oligonucleotide-conjugated antibodies (PLA probes: anti-mouse MINUS and anti-rabbit PLUS) were used to detect primary antibodies. Fluorescence signal was detected using the Zeiss AxioObserver Z1 microscope described above. Images were collected in optical planes with a z-axis of approximately 0.5 μm per step. Stacks of images were combined in Fiji into a single-intensity projection and used to quantify PLA puncta.

## Fluorescence recovery after photobleaching

Cells were bathed in a modified Krebs-Ringer buffer (KRB) solution containing (in mM): 146 NaCl, 4.7 KCl, 2.5 CaCl$_2$, 0.6 MgSO$_4$, 1.6 NaHCO$_3$, 0.15 NaH$_2$PO$_4$, 8 glucose, 20 HEPES, pH 7.4, ~330 mOsm, at 37 °C and imaged on a Zeiss LSM 880 Airyscan confocal microscope using a Plan-Apo 63×/1.4 Oil objective. The fluorescent proteins tagBFP, eGFP, and mScarlet were excited with laser lines 405 at 3%, 488 at 4%, and 543 at 10%, respectively, and emission was detected with PMT detectors with ranges 410–481 nm, 493–553 nm, and 554–696 nm, respectively. Images were taken of 512 × 512 pixel size with a zoom of 2 times and a pinhole of 1 AU such that optical sectioning was 0.9 μm. A circular region of interest (ROI) with 2 μm diameter was drawn over a cluster for stimulation. Photobleaching was achieved by exposing the ROI to lasers 405 and 488 or lasers 488 and 543 at 100% power in HEK293T cells and neurons, respectively, for 200 iterations and applying the zoom bleach setting. FRAP series consisted of 180 frames with 5 frames acquired before photobleaching and 175 frames after photobleaching, with an interval of 1 Hz. For experiments with DsRed-Monomer, only the circular ROI was acquired during the FRAP series which consisted of 500 frames with 10 frames acquired before photobleaching with an interval of 20 Hz.

## Electron microscopy

**Immunoperoxidase-EM.** All rat handling and sample preparation procedures for immunoperoxidase immunoelectron microscopy were performed at the Oregon Health and Science University and were conducted under regulatory guidelines of the institution. Male Sprague-Dawley rats ($n = 2$) were perfused transcardially with 10 mL of heparinized saline followed by 600 mL of 4% formaldehyde (freshly prepared from paraformaldehyde). Brains were postfixed for 30 min in the same fixative then placed in 0.1 M PB. The hippocampal region of the brain was sectioned (40 μm) on a vibrating microtome then processed for electron microscopy immunohistochemical labeling. Tissue sections were first incubated in 1% sodium borohydride solution for 30 min followed by 30 min in 1% BSA. Tissue sections were then incubated for 72 h (3 nights) at 4 °C in a primary antibody solution of mouse anti-SPHKAP IgG1 mAb L131/17. The primary antibody solution contained 0.01% Triton X-100 (Sigma-Aldrich, St. Louis, MO) and 0.1% bovine serum albumin (BSA, Sigma-Aldrich) in 0.1 M Tris-Saline. Following the 3 day incubation, tissue sections were incubated in secondary antibody solution for two hours at room temperature. The secondary antibody solution contained biotinylated goat anti-mouse (1:400, Vector Labs Cat # BA-9200-1.5) and 0.1% BSA (Sigma-Aldrich) in 0.1 M Tris-Saline. Tissue sections were then rinsed and incubated for 30 min in ABC solution (Vectastain, Vector Labs Cat# PK-6100) made in 0.1 M Tris-Saline. Following ABC incubation, sections were then incubated for 3 min and 30 seconds in a DAB solution containing 11 mg of 3,3'-Diaminobenzine, 50 mL of 0.1 M Tris-Saline and 5 μL of 30% hydrogen peroxide. Tissue sections were then osmicated using a solution of 1% OsO$_4$ in dH2O. Sections were then dehydrated through a series of graded ethanol dilutions (50%, 70%, 90% and 100%) for 5 min each. Dehydrated sections were placed in a 1:1 mixture of 100% ethanol to Epon resin (Electron Microscopy Sciences Cat# 14120) for 30 min before being placed in pure Epon resin (EMS) overnight for infiltration. Resin infiltrated sections were then flat embedded between Aclar sheets (Electron Microscopy Sciences Cat# 50425-10) cured at 60 °C overnight then glued onto resin blocks. The CA1 region of the hippocampus was identified, the section trimmed, and ultra-thin sections (60–70 nm) were placed on T400-Cu grids (Electron Microscopy Sciences Cat# EMS400CU). Grids were then counterstained using Uranyl Acetate at 5% and Reynolds Lead Citrate before being imaged on an FEI Tecnai T12 Transmission Electron Microscope interfaced to an AMT digital camera.

**Immunogold-EM.** Pre-embedding immunogold electron microscopy was performed as described previously[62]. All mouse handling and sample preparation for pre-embedding immunoelectron microscopy was performed at the Institute of Science and Technology Austria and were conducted under the license approved by the Austrian Federal Ministry of Science and Research and the Austrian and EU animal laws. Deeply anesthetized mice were transcardially perfused with 25 mM PBS, pH 7.4, for 1 min, followed by a fixative containing 4% formaldehyde (freshly prepared from paraformaldehyde), 0.05% glutaraldehyde, and 15% saturated picric acid made up in 0.1 M phosphate buffer (PB), pH 7.4, for 12 min. After perfusion, brains were quickly removed from the skull, washed briefly with PB, and coronally sectioned at 50 μm with a microslicer (Pro-7 linear slicer, Dosaka). Sections were rinsed several times in PB, freeze-thawed, quenched with 50 mM glycine in Tris-buffered saline (TBS) pH 7.4 at room temperature for 1 h, blocked in 10% NGS and 2% bovine serum albumin in TBS for 1 h, and then incubated in primary antibodies made up in TBS containing 1% NGS overnight at 4 °C. For single labeling for SHPKAP, mouse monoclonal anti-SPHKAP IgG1 antibody L131/17 (at 2 μg/mL) was used. For the double labeling, the SHPKAP antibody was combined with rabbit antibodies for PKARI (1 μg/mL, Cell Signaling Technology). After washing, the sections were incubated with 1.4 nm gold-coupled anti-mouse secondary antibody (Nanoprobes Cat# 2001) diluted in TBS at a ratio of 1:100. After washing, the sections were postfixed in 1% glutaraldehyde for 10 min, followed by silver enhancement of the immunogold particles using a HQ silver EM intensification kit (Nanoprobes Cat# #2012). Sections for the double labeling were washed in TBS and incubated with biotinylated goat anti-rabbit IgG antibody (1:200, Vector Labs Cat# BA-1000) at room temperature overnight. The sections were washed in TBS and then reacted with an avidin-biotinylated horseradish peroxidase complex (1:100 Vectastain Elite, Vector Labs Cat# PK-6200) made up in TBS for 2 h. After three washes in TBS and one in 50 mm Tris-HCl buffer (TB), pH 7.4, the peroxidase activity was visualized by incubating sections in TB containing 0.025% 3,3'-diaminobenzidine tetrahydrochloride (Dojindo Molecular Technologies) and 0.003% hydrogen peroxide. Sections for both single and double labeling were then postfixed with 1% osmium tetroxide for 20 min, en bloc counterstained with 1% uranyl acetate for 30 min, and dehydrated in graded ethanol series followed by propylene oxide. The sections were infiltrated overnight at room temperature in Durcapan resin (Sigma-Aldrich Cat# 44611) and transferred to glass slides for flat embedding. After resin curing at 60 °C for 48 h, the

trimmed tissues from the CA1 area of the hippocampus were re-embedded in Durcapan resin blocks for ultrathin sectioning. Serial 70-nm-thick sections were cut from the surface (within 3 μm depth) of the samples and were collected on pioloform-coated single-slot copper grids. Images were captured with a CCD camera (Veleta, Olympus SIS) connected to a Tecnai 12 transmission electron microscope (FEI).

**Thin-section TEM on HEK293T cells.** HEK293T cells cultured in 35 mm dishes (Corning Cat# 353001) were co-transfected with SPHKAP-mScarlet or SPHKAP-C-mScarlet and RIα-GFP. 48 h after transfection, the media was removed, and the cells were scraped from the dish in 1 mL of PBS. The cell suspension was pelleted in a 1.5 mL microcentrifuge tube by centrifugation at 150×*g* for 5 min. The cell pellet was then fixed in 2.5% formaldehyde (Electron Microscopy Sciences Cat# 15710)/2.5% glutaraldehyde (Ted Pella, Inc, Cat# 18426) in PBS, pH 7.4 overnight. The fixative was then removed, and the cells were rinsed in PBS, followed by a secondary fix in 1% osmium tetroxide/1.5% potassium ferrocyanide and for 1 h. The cells were then rinsed three times in cold ddH$_2$O. Cells were then dehydrated with successive 10 min incubations in 30%, 50%, 70%, 95%, and three washes in 100% ethanol, followed by incubation in propylene oxide 2 × 10 min. The dehydrated cell pellet was then pre-infiltrated with half propylene oxide/ half resin overnight, then infiltrated with 100% resin (dodecenyl succinic anhydride, Araldite 6005, Epon 812, dibutyl phthalate, benzyldimethylamine) for 5 h. The resin was then removed and replaced with fresh resin and allowed to polymerize at 70 °C overnight. Approximately 100 nm-thick sections from the resin blocks were sectioned on a Leica EM UC6 ultramicrotome and collected on copper grids. Grids were stained with 4% aqueous uranyl acetate and 0.1% lead citrate in 0.1 N NaOH and imaged in a FEI Talos L120C TEM.

**Stimulation and pharmacological treatment of cells**
**Depolarization-induced activation of CREB and c-Fos.** Depolarization-induced activation of transcription factors in control (scr shRNA) or SPHKAP KD (SPHKAP shRNA) DIV15 cultured hippocampal neurons was performed as previously described[11]. The culture medium was replaced with the same modified KRB described above and supplemented with 1 μM TTX (Alomone Cat# T-550), 10 μM AP5 (Alomone Cat # D-145), and 50 μM CNQX (Alomone Cat# C-141) to block intrinsic neuronal activity for 2 h at 37 °C. Cells were then stimulated with 45 mM K$^+$-containing KRB at 37 °C (with TTX, AP5, and CNQX present and NaCl reduced to maintain osmolarity) for 90 seconds, then returned to neuronal culture medium for 3 h (for immunolabeling and evaluation of c-Fos) or immediately fixed (for immunolabeling and evaluation of phospho-Ser133 CREB).

**Assessment of PKA activation in fixed neurons.** DIV15-21 hippocampal neurons were incubated with the same KRB solution supplemented with synaptic blockers detailed above for 2 h at 37 °C followed by depolarization with elevated K$^+$-containing KRB solutions (10, 20, 30, or 45 mM K$^+$; with TTX, AP5, and CNQX present and NaCl reduced to maintain osmolarity) for 90 seconds, then immediately fixed and processed for immunolabeling. For experiments including TAT peptides, cells were incubated overnight with TAT-Scr or TAT-CCAD[11] diluted to 1 μM in culture medium and incubated and stimulated with KRB solutions containing 1 μM TAT peptide. For stimulation of neurons with isoproterenol (ISO, Sigma-Aldrich Cat# I5627), ISO was added to KRB solution from a freshly prepared stock solution immediately prior to stimulation of neurons. In some experiments, H89 (Cayman Chemical Cat# 10010556) or nimodipine (Alomone Cat# N-150) were included at the indicated concentrations.

**Hexanediol and forskolin/IBMX treatment.** HEK293T cells and DIV20-21 hippocampal neurons were incubated in pre-warmed KRB supplemented with 3% (wt/vol) 1,6-hexanediol (Sigma-Aldrich Cat#

240117) or 20 μM forskolin (Cayman Chemical Cat# 11018) and 100 μM IBMX (Sigma-Aldrich Cat# I7018) (neurons only) at 37 °C for 5 min then immediately fixed as described above and processed for immunolabeling.

**TAT-FFAT peptides.** The FFAT motif sequence used for the cell-penetrating TAT-FFAT peptides was derived from the sequence of rat ORP1 (NCBI Reference Sequence: NP_742020.1). Three variants were synthesized by GenScript at over 95% purity: TAT-FFAT-HA (472-481) (GRKKRRQRRRPQSEDEFYDALSYPYDVPDYA; "472-481" corresponding to amino acid residue numbers in rat ORP1); TAT-FFAT-HA (472-481) scr (GRKKRRQRRRAYESDQLPSFDEYPYDVPDYA; scrambled control of TAT-FFAT-HA (472-481)); and TAT-FFAT-HA (465–494) (GRKKRRQRRRPQSPPVSILSEDEFYDALSGSESEGSLTCLEAYPYDVPDYA; "465–494" corresponding to amino acid residue numbers in rat ORP1). Peptides were prepared in aliquots with ultrapure water. Cells were incubated overnight with TAT peptides diluted to 5 μM in culture medium.

**Ca$^{2+}$ and PKA biosensor imaging.** To load shRNA-transfected DIV14-15 hippocampal neurons with Cal-590 AM (AAT Bioquest Cat# 20510), cells were first incubated in regular culture medium to which had been added 1.5 μM Cal-590 AM for 45 min at 37 °C. Dye-containing medium was then aspirated, followed by two washes in KRB which had been warmed to 37 °C. For Cal-590-loaded neurons and for neurons expressing ExRai-AKAR2 or jRGECO1a, cells were incubated in KRB supplemented with TTX, AP5, and CNQX for 30 min at 37 °C prior to imaging, imaged in this same solution, and stimulated with 45 mM K$^+$-containing KRB at 37 °C (with TTX, AP5, and CNQX present and NaCl reduced to maintain osmolarity). Images were acquired using a Zeiss LSM880 confocal laser scanning microscope using a 20×/0.8 NA Plan-Apochromat objective. Acquired image stacks were processed and analyzed using Fiji.

**Image analysis**
Morphological and colocalization analyses of fluorescent and immunolabeled proteins were performed using Fiji. Custom macros were used to automate analyses and minimize experimenter bias. For analysis of SPHKAP, RI, and AKAP11 cluster size, images were subjected to "rolling ball" background subtraction and subsequently converted into a binary mask by thresholding. For the colocalization analyses, an ROI was drawn around the cell body and Pearson's correlation coefficient (PCC) values were collected using the Coloc2 plugin. All intensity measurements reported in line scans are normalized to the maximum intensity measurement. Immunolabeling intensity and PCC values within hippocampal area CA1 somas imaged in mouse brain sections was measured using an annular ROI with a 1.75 μm-wide band generated with a custom macro using Fiji (NIH) software. The area of the thresholded SPHKAP clusters was measured using the "Analyze particles" function, measuring all particles larger than 1 pixel. Measurement of VAPA/B immunofluorescence signal intensity on or off SPHKAP clusters was performed by using a thresholded SPHKAP cluster image (generated as above) to identify SPHKAP clusters; on-cluster VAPA/B immunofluorescence corresponded to the signal overlapping with thresholded SPHKAP clusters, whereas off-cluster VAPA/B immunofluorescence was all signal detected exclusive of thresholded SPHKAP clusters.

SPHKAP-mScarlet and RIα-GFP clustering was measured by quantifying the coefficient of variation (CV) of the fluorescence intensity (fluorescence s.d./mean s.d.) in each cell, with a greater CV indicative of a higher degree of protein clustering[63]. Nearest neighbor distances (NNDs) of immunolabeled somatic SPHKAP and Kv2.1, RI, VAPA/B, Cav1.2, or PKA-C clusters in cultured hippocampal neurons were calculated by first converting each channel to a binarized mask and obtaining the centroid coordinates of the labeled clusters in the

thresholded images. NNDs were then calculated from these coordinates using R package 'spatstat' (version 2.3–4). Overlap between SPHKAP and Kv2.1 immunolabeling was determined by multiplying the thresholded super-resolution localization map images for SPHKAP and Kv2.1, quantifying the area of the clusters in the resultant image, and reporting as a percentage of the total area of binarized Kv2.1 clusters. For presentation, images were exported as TIFFs and linearly scaled for min/max intensity and flattened as RGB TIFFs in Photoshop (Adobe).

### Statistical analysis

For all data sets presented in this study for which statistical analyses were performed, measurements were imported into GraphPad Prism for statistical analysis and presentation. Reported values are mean ± s.e.m. Paired data sets were compared using Student's $t$-test if the data passed a normality test; a non-parametric test was used otherwise. Exact $P$-values and the statistical test used are reported in each figure or figure legend. For all experiments at least two independent cultures were used for experimentation; the number of samples analyzed is stated in each figure or figure legend.

### Reporting summary

Further information on research design is available in the Nature Portfolio Reporting Summary linked to this article.

## Data availability

All data supporting the findings of this work can be found in the source data file and Supplementary Information; additional raw data will be made available upon request. The proteomics data generated in this study have been deposited to the ProteomeXchange Consortium via the PRIDE partner repository with the dataset identifier PXD044574. Source data are provided with this paper.

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

## Acknowledgements

We thank Kayla Templeton and Peter Turcanu for technical assistance, Michelle Salemi for assistance with LC-MS data acquisition and analysis, Dr. Belvin Gong for advice on monoclonal antibody generation, Drs. Maria Casas Prat and Eamonn Dickson for assistance with super-resolution TIRF microscopy, Dr. Oscar Cerda for assistance with the design of TAT-FFAT peptides, Dr. Fernando Santana for helpful discussions, and Dr. Jodi Nunnari for a careful reading of our manuscript. We also thank Dr. Alan Howe, Dr. Sohum Mehta, and Dr. Jin Zhang for providing plasmids used in this study. This project was funded by NIH Grants R01NS114210 and R21NS101648 (J.S.T.), and F32NS108519 (N.C.V.).

## Author contributions

N.C.V. and J.S.T. conceived the project. N.C.V., L.R.-S, M.K., S.A.A., R.S., and J.S.T. designed the experiments. N.C.V., L.R.-S, M.K., D v.d.L., P.B., O.A.M., J.C., and E.L.M performed the experiments. N.C.V., L.R.-S, M.K., S.A.A., R.S., and J.S.T. analyzed data. N.C.V. and J.S.T. coordinated the study and provided guidance. N.C.V., L.R.-S, S.A.A., R.S., and J.S.T. wrote the paper. All authors discussed the results and approved the final version of the manuscript.

## Competing interests

The authors declare no competing interests.
