## [Peer Review File · Nature Communications]

Neuronal ER-plasma membrane junctions couple excitation to Ca²⁺-activated PKA signalingREVIEWER COMMENTS

Reviewer #1 (Remarks to the Author):

In this paper, Vierra et al., report that type I PKA and Kv2.1 form a signaling complex through the PKA anchoring protein SPHKAP at ER-PM junctions. They show that Kv2 channels are necessary for the clustering of SPHKAP. They also claim that a SPHKAP-RI-VAP complex controls the arrangement of the ER. This study also investigates the roles of the protein complex and finds that SPHKAP KD impairs KCl-mediated depolarization-induced activation of CREB and c-Fos. Finally, PKA activity upon KCl treatment is investigated in SPHKAP KD neurons. While research on the PKA signaling pathway in neurons has been extensively conducted, the topic remains important. The notion that the initiation of signaling occurs at the ER-PM junction is an attractive model. However, the current data presented in the manuscript are not sufficient to support the claim that a signalosome, including PKA, localizes at the junction. Instead, the data suggest that the signalosome may be located away from the PM. Additionally, the manuscript is written in a somewhat confusing manner, as it discusses SPHKAP's localization in the cytosolic space between stacked ER in one figure, while focusing on its localization at ER-PM junctions in other figures.

Major points

- 1- The spatial resolution of subcellular localization analyses is not sufficient to conclude that "SPHKAP anchors type I PKA at ER-PM junctions". Fig. 1a, obtained with a 10x/0.5 NA lens, does not provide enough resolution to judge subcellular level localization. To claim "with SPHKAP's subcellular localization resembling Kv2.1's somatodendritic distribution", investigations with much higher microscopy or biochemical experiments using fractionation, for example, should be performed.
- 2- The result of immunoprecipitation experiment using DSP crosslinking does not necessarily mean that SPHKAP localizes in close proximity to Kv2.1, as the proteins may be associated through formation of a large protein complex or membrane structures. The proximity labeling assay could be performed to address this point.
- 3- Along the same vein, the entire proteomics data should be shared in order to give a rough idea about the Kv2.1 complex.
- 4- It is claimed that "SPHKAP and RI exhibited a strikingly punctate distribution in hippocampal pyramidal cell somata" (Line 79.). Since axons and dendrites are highly packed around soma in the hippocampus, it is hard to determine if the signals are from somata or from other compartments. Co-labeling with markers for soma, dendrites, and axon would make the point clear.

5- The super resolution imaging (Fig. 1f) is an adequate approach for analyzing the colocalization. However, the overlaps of the protein pairs could be resulted from coincidental co-localization. In particular, the overlap between Kv2.1 and SPHKAP, which is a novel finding, needs to be statistically analyzed to verify the results.

6- The immunoEM analysis is potentially a key experiment to determine the localization of SPHKAP. However, the DAB enhanced signals in Fig. 1g are too diffused for determining the actual location of SPHKAP. Also, the labeling with gold particles in Fig. 1h and 1i clearly shows that SPHKAP locates on the other side of ER facing the PM, as described in the diagram (Fig. 2h). These data do not align with the authors statement “type I PKA is concentrated at ER-PM junctions”.

7- The experiments using shRNA targeted to SPHKAP are critical experiments in determining the roles of SPHKAP. To exclude the possibility of off-target effects, multiple shRNA should be tested. Indeed, the authors showed there are three more efficient shRNA sequences (Extended data 8). For the analysis of PKA-C localization (Fig 3e.f), the authors should examined whether SPHKAP knock down affects the expression level of PKA-C.

8- Their claim made in lines 175-178 is an overstatement.

9- It should be explained why the quantification of “depolarization-induced LTCC-dependent activation of somatic PKA in neurons” is an investigation for “whether type I PKA localization near LTCCs impacts its function”.

10- In Fig. 4C, the association of PKA-C with SPHKAP should be statistically analyzed.

Minor point

A scale bar for Fig. 1g (i) is probably incorrect.

Reviewer #2 (Remarks to the Author):

The manuscript by Vierra et al., ‘ER-PM junctions couple excitation to Ca²⁺-activated PKA signaling’ provide new insights into subcellular compartmentalization of soma-enriched type I PKA at ER-PM junctions. The authors identify SPHKAP (type I PKA anchors) as a protein interactor of KV2.1 channels.

SPHKAP anchors type I PKA at ER-PM junctions and creates a PKA dependent signaling pathway to support Ca²⁺ influx and excitation-transcription coupling. Overall, the proposed model is very attractive and it shows for the first time a PKA signalosome associated to ER-PM, with an essential physiological role. However, some conclusions are overstated. Also, it is unclear how the formation of SPHKAP-RI condensates contributes to the functional role of ER-PM contacts in neurons. In addition, some experiments are not clearly explained in the main document. See below specific concerns.

Major Points:

1. The authors show in Fig 1 and Fig S1 the subcellular localization of SPHKAP, Kv2 channels and RI subunit in mouse brain sections by confocal, TIRF and EM. Similar results in cultured hippocampal neurons (Fig S2). The conclusion of this data is that SPHKAP anchors type I PKA at ER-PM junctions. Although intensity profile lines showing overlap in specific regions, and correlation graph are presented (Fig 1C, Fig S1f), there is not quantitative evidence of how many of the Kv2 channel, SPHKAP and RI clusters overlap with each other. What is the percentage of Kv2 channel clusters associated to SPHKAP, RI and VAP? For instance, in Fig S2, many clusters of KV2.1 are observed, and only few are overlapping with RI and SHKAP. Data in Fig S2 should be quantified. The authors state that around a 47% of SHKAP is present in ER cistern stacks adjacent to PM, and 53% in ER cistern stacks no adjacent to PM. Is this consistent with number of SHKAP clusters associated to Kv2?

2. In the presented data there is no evidence showing that SHKAP 'anchors' type I PKA at ER-PM (title section and Fig 1). Data shows co-distribution of Kv2, SHKAP and RI. Co-staining with VAP is also shown (TIRF, Fig. 1f; Fig S3). Here, it is not evaluated whether SHKAP can or not anchor type I PKA to ER-PM junctions. For this claim, the use of a shRNA against SHKAP or dominant negative to interfere with Kv2 should be provided.

3. The role of Kv2 in SPHKAP clustering is also examined in Kv2 single and double KOs. Authors state 'this suggest that Kv2 channels are determinant of SPHKAP-RI subcellular distribution but not expression'. Does Kv2 KO impair ER-PM contact or just clustering of SPKAP, RI? It looks like Kv2 DKO drastically impairs SHKAP clustering. Considering that around 53% of SHKAP present in stacked ER cisterns is not in contact with the PM, could Kv2 KO indirectly affect distribution of SHKAP? This should be discussed.

4. In Fig S3, VAPA/B co-distribution with SPHKAP and Kv2 is examined. Only representative images are shown. The authors state that VAP density co-varied with SPHKAP clusters. This is not clear from images and should be quantified in terms of number of clusters co-localizing with each other.

5. Authors moved to heterologous expression in HEK cells to study SPHKAP-RI inter-dependency in cluster formation and to evaluate the role in ER remodeling (Fig 2, Fig S5). The clusters observed for

mScarlet-tagged SPHKAP could be just an artifact of the tag aggregation at high expression levels. It is unclear whether images of co-expressed proteins (and deletion constructs) are examined at low or high expression levels. Deletion constructs used are not clearly explained in main text. To confirm reciprocal co-clustering, the authors could use a smaller tag for SPHKAP (e.g., HA tag) However, it could be better to show the effect of shRNA for SPHKAP and RI, and rescue experiments with deletion constructs, or use a Dominant negative approach to specifically compete SPHKAP-RI association in cultured neurons. This will more clearly show the interdependency between them in cluster formation in neurons.

6. The analysis of liquid-liquid phase separation properties of these SPHKAP-RI clusters is conceptually unclear. Is there any disordered region in these proteins that could suggest a role in maintaining phase separation of clusters in neurons? Hexanediol treatment has just little effect on SPHKAP-RI cluster formation. FRAP experiments also show little mobility within assemblies. Droplet fusion is another property of phase separated assemblies, but this is not examined. The current data suggest that they are not liquid-liquid phase separated structures. The experiment with GFP, to study permeability of assemblies to molecules is not properly performed. Resolution of the experiment is not adequate to test this. At least FRAP for GFP should be performed to study GFP diffusion outside and within assemblies. The data does not add much understanding to the proposed model. Also, there are experiments in Fig S5, which are not explained, such as the use of forskolin/BMX, H89. I would suggest to remove all of this from the manuscript.

7. The role of these 'condensates' (I would not call them condensates) in ER remodeling is not well examined. Distribution of VAP is examined in the presence of overexpressed SPHKAP-scarlet and RI-GFP in neurons (Fig S7). Expressed RI-GFP does not colocalize with VAPA/B clusters and total VAPA/B level appears to be reduced in this condition. SPHKAP-scarlet distribution is not clear, separated channels should have been provided not only for VAP. Compared to endogenous SPHKAP (Fig S2), the overexpressed protein label less/smaller SPHAP clusters. Changes in VAP looks modest, and perhaps a reduction in number and increase in size. Does clustering quantification account by these two parameters? Same experiment was performed in HEK cells (Fig 2g, Fig S7). In addition, EM of HEK cells co-expressing proteins increases OSER compared to non-transfected cells (Fig 2d). Are these OSER associated to the PM? Is expression of SPHKAP and mutants low or high expression? This is unclear. The conclusion that SPHKAP-RI assemblies remodel ER (title S7) is overstated, here only the distribution of VAP is examined. A clearer experiment to demonstrate a role of SPHKAP-RI complex in VAP distribution and ER remodeling would be knocking down SPHKAP and rescue with FL and mutants in cultured neurons and analyze VAP distribution and ER remodeling using a general ER marker such KDEL, Sec61b. Does SPHKAP KD impair stacked ER cisterns? Is ER-PM interaction impaired in this condition? These experiments could strengthen your proposed model presented in Fig 2h.

8. It is not clear why the authors quantify distance between SPHKAP and PKA-C respect to Ca1.2 in Fig 3c-d. This is just confocal microscopy, there is not enough spatial resolution to examine the close proximity of these proteins. For this experiment better to show colocalization analysis or % of cluster colocalization. Although overlap between SPHKAP and PKA-C is clear, the co-distribution of SPHAP with

KV2.1, Ca1.2 and RyR is not clear. PLA is a nice way to study the interaction at the nanoscale resolution, but why there are some puncta outside the soma? Is SPHAP distributed also along dendrites? (Fig 3e). Since the authors have a good shRNA against SPHKAP, it would be good to study the requirement of SPHKAP in the distribution of markers shown in Fig 3A. Is SPKAP playing a structural role in connecting ER-PM to ca²⁺ channels and PKA signaling? Or there is only a functional role as shown in Fig 3g-k?

Minor comments:

1. Figure 2a is not clearly explained in result section.
2. In extended Figure 5 and 6, authors are checking whether SPHKAP clusters have LLPS properties. There is not reference to Forskolin/ IBMX and H89 experiment.
3. There are no scale bars for extended figure 5d
4. It should be explained why PLA assay was used in the result section

Reviewer #3 (Remarks to the Author):

This is a substantial investigation that includes a number of interesting and novel observations. Notably, the authors show that: (i) SPHKAP-PKA type I associates with Kv2.1 channels resulting in localisation of the complex at ER-PM junctions. This part of the study is robust and very firmly established using multiple complementary techniques including beautiful EM imaging (ii) They show that interaction between PKA RI subunits and SPHKAP supports clustering of both proteins and is linked to generation of organised smooth ER (iii) They show that SPHKAP is required for Ca²⁺ elevation triggered by potassium-induced activation of voltage-gated calcium channels (iv) they show that SPHKAP is necessary for activation of PKA that results from Ca²⁺ elevation following activation of these channels.

Overall, this is an impressive body of work, and will interest researchers in EM-PM junctions, and in localised cAMP/PKA signalling. Nevertheless, I was not convinced of the logic of the mechanism presented in Fig. 2h, and there are also some uncertainties regarding exactly how SPHKAP is supporting PKA activation following activation of LTCCs (Figure 4).

Specific comments:

1. With respect to the relative importance of different PKA regulatory subunit isoforms. It would be worth comparing their findings with studies that have imaged PKA subunit distribution in the hippocampus (Ilouz et al., eLife, 2017) and quantified the abundance of PKA subunits in different brain regions (Walker-Gray, PNAS, 2017). Potentially, Ilouz et al., indicates that CA1 neurons are somewhat unusual in expressing Ribeta at high levels in the cell body.
2. Furthermore – as summarised by Kirschner et al., Mouse models of altered protein kinase A signaling – there is evidence that Ribeta is particularly important in the hippocampus
3. Line 3-76. SPHKAP distribution is exclusively somatic rather than somatodendritic judging by the supplementary images
4. In figure 2, and on lines 116-129, it could be emphasised that – unusually – SPHKAP has two RI anchoring sites. So it is immediately apparent how SPHKAP could support RI clustering (but not so the converse)
5. I was confused as to how figures 2c and extended figure 5a relate – the numbers are different whereas the conditions look the same?
6. It is not clear why the D/D-linker version of RIalpha is able to support clustering of SPHKAP... Zhang et al., Cell, 2020 (paper on RI phase separation) clearly shows that the cyclic nucleotide binding domains of RI are required for phase separation of RI. How do the authors reconcile these observations?
7. Related to this point, in the summary figure presented in fig. 2h, in the cartoon RI is presented as bridging two SPHKAP molecules. This does not seem logical given what is known about the structural basis of RI-SPHKAP interaction. Means et al., 2011 showed that RI binds to amphipathic anchoring helices presented by SPHKAP via the RI dimerization and docking domain – ie there is no way for an RI dimer to bridge two copies of SPHKAP given that dimerization and anchoring to a single site occur within the same small domain. Potentially phase separation-type interaction of RI dimers bound to different SPHKAP polypeptides could support SPHKAP clustering, However, the hexane diol experiments presented in extended figure 5 do not support an important role for phase separation in the clustering process. I thought this was the least convincing and most confusing aspect of the study.

8. In figure 3c & d – what is the average distance to a randomly selected point in the soma? (ie the average distance will naturally reduce as the density of SPHKAP/PKA-C points increases)

9. Regarding reference 33 (Dell'Acqua study) – the authors state that 'consistent with the observation that direct inhibition of PKA or its uncoupling from AKAPs suppresses LTCC activity...' I feel this is a little misleading since the work from Dell'Acqua/Sather highlights a critical role for AKAP79-type II PKA complexes. Ie, the notion that SPHKAP-type I PKA is key to PKA activation downstream of LTCCs challenges these studies rather than being consistent with them. The authors should acknowledge this difference and comment on how they view the relative importance of these two AKAPs in responding to Ca²⁺ influx through LTCCs.

10. In terms of interpreting the results of figure 4 – this is an interesting finding although some uncertainties remain around the mechanism. Specifically:

(i) Is Ca²⁺ release through ryanodine receptors required for the elevation in Ca²⁺ seen in e.g. Fig 3g?

(ii) Do the authors think that C subunits binds to SPHKAP-RI are essential for the PKA activity measured in the experiments presented in figure 4? Note, they detected robust C subunit co-precipitation with Kv2.1 channels by MS. Conceivably, however, the complex could serve a purely structural role given that SPHKAP knockdown prevents LTCC-triggered Ca²⁺ elevation (Fig 3g).

Regarding these points, ideally the authors could perform additional experiments with RyR blockers, and potentially using a knockdown-replacement approach to substitute in mutated versions of RI subunits that are not able to interact with PKA C subunits. If this is not feasible then they should more clearly highlight remaining uncertainties in the mechanism.

REVIEWER COMMENTS

Reviewer #1 (Remarks to the Author):

In this paper, Vierra et al., report that type I PKA and Kv2.1 form a signaling complex through the PKA anchoring protein SPHKAP at ER-PM junctions. They show that Kv2 channels are necessary for the clustering of SPHKAP. They also claim that a SPHKAP-RI-VAP complex controls the arrangement of the ER. This study also investigates the roles of the protein complex and finds that SPHKAP KD impaires KCl-mediated depolarization-induced activation of CREB and c-Fos. Finally, PKA activity upon KCl treatment is investigated in SPHKAP KD neurons. While research on the PKA signaling pathway in neurons has been extensively conducted, the topic remains important. The notion that the initiation of signaling occurs at the ER-PM junction is an attractive model. However, the current data presented in the manuscript are not sufficient to support the claim that a signalosome, including PKA, localizes at the junction. Instead, the data suggest that the signalosome may be located away from the PM. Additionally, the manuscript is written in a somewhat confusing manner, as it discusses SPHKAP's localization in the cytosolic space between stacked ER in one figure, while focusing on its localization at ER-PM junctions in other figures.

We thank the reviewer for noting that the relationship between PKA signaling in neurons and the "initiation of signaling occur[ing] at the ER-PM junction" is an "attractive model." We also thank the reviewer for their careful review and their thoughtful feedback. We have performed extensive new experiments that together result in a total of six new figures (two main and four supplemental). We have also made numerous revisions that we feel have substantively strengthened our manuscript and provided additional support for our overall conclusions. We thank the reviewer for their comments that led to this outcome.

As to the primary concern raised by the reviewer here, we acknowledge that the vocabulary used in the original manuscript was imprecise and inconsistent in blurring the important distinction between the ER-PM junction itself (of the type found in many other eukaryotic cells) and the ER-PM junctions associated with subsurface ER cisternae that to our knowledge are unique to neuronal somata. We are now consistent in our use of the "ER-PM junction" when referring to the junction itself, and "subsurface ER cisternae" when referring to the cisternae themselves. We have also defined the overall domain on which our study focuses as the "ER-PM junctional domain", which comprises the ER-PM junction itself as well as the associated subsurface ER cisternae. We explicitly define this terminology at the outset and have revised the manuscript throughout to be consistently clear when referring to one or the other individual component of the domain, or the overall domain itself. We have also incorporated substantial new results acquired to address this and the other reviewer's questions. We address specific points below.

Major points

1- The spatial resolution of subcellular localization analyses is not sufficient to conclude that "SPHKAP anchors type I PKA at ER-PM junctions". Fig. 1a, obtained with a 10×/0.5 NA lens, does not provide enough resolution to judge subcellular level localization. To claim "with SPHKAP's subcellular localization resembling Kv2.1's somatodendritic distribution", investigations with much higher microscopy or biochemical experiments using fractionation, for example, should be performed.

We concur that our conventional light microscopy images provide insufficient resolution to unequivocally determine the subcellular structures with which SPHKAP-type I PKA assemblies associate. We replaced the previous statement regarding Fig. 1a (now supplemental Figure 1a) "SPHKAP's subcellular localization resembles Kv2.1's somatodendritic distribution," with the revised statement "Indeed, multiplex labeling of brain sections for Kv2.1, SPHKAP, and PKA-RI revealed an overall similarity in the distribution of all three proteins on neuronal somata and proximal dendrites in the hippocampal formation". (lines 94-96).

We also thank the reviewer for noting that additional experiments are necessary to define the subcellular localizations of Kv2.1, SPHKAP and type I PKA. We have now performed these experiments. We describe the extensive new experiments performed to address this point in detail in the manuscript and in our responses below.

2- The result of immunoprecipitation experiment using DSP crosslinking does not necessarily mean that SPHKAP localizes in close proximity to Kv2.1, as the proteins may be associated through formation of a large protein complex or membrane structures. The proximity labeling assay could be performed to address this point.

We thank the reviewer for raising this important point. We appreciate the reviewer's suggestion of performing proximity ligation assay (PLA) to assess the spatial relationship between SPHKAP and Kv2.1. We have now performed extensive PLA experiments in cultured neurons, including those subjected to shRNA-mediated knockdown of SPHKAP, the results of which are shown in Fig. 2g of the revised manuscript. We also included new PLA experiments that assess in parallel the relationship between SPHKAP and VAPs (Fig. 2h) as well as RyRs and RI (Fig. 5b). These experiments show that control neurons displayed robust PLA signal between Kv2.1 and SPHKAP, whereas neurons treated with either of two distinct anti-SPHKAP shRNAs displayed little to no PLA signal between SPHKAP and Kv2.1. Similarly, we found that SPHKAP and VAP yielded prominent PLA signal in control neurons, as did RyR and RI, and that PLA signal for these pairs was abolished in SPHKAP knockdown neurons. Combined with the other results presented in this revised manuscript, these new data further strengthen our conclusion that the SPHKAP-type I PKA signalosome is tightly associated with ER cisternae near Kv2.1-mediated ER-PM junctions. We thank the reviewer for their suggestion.

3- Along the same vein, the entire proteomics data should be shared in order to give a rough idea about the Kv2.1 complex.

We have included a new Supplemental Table 2 that contains a list of the proteins specifically purifying with Kv2.1 that have been identified in these experiments. We also add to this new Supplemental Table 2 unpublished data listing the top proteins co-purifying with the axonal Kv1.2 K⁺ channel in immunopurifications performed in parallel with those for Kv2.1. The lack of overlap between these two extensive datasets further illustrates the specificity of these separate productive immunopurifications that each yielded a large set of mutually exclusive proteins. There are two small sets of proteins copurifying with Kv2.1 that are not listed in this table. One is a set of electrically silent K⁺ channel subunits that are novel components of the Kv2 channel complex itself, and that are the focus of a distinct manuscript with collaborators. The other are a small set of proteins of unknown function that form the basis of future research projects that the first author (NCV) intends to pursue in his new independent laboratory at the University of Utah beginning in January 2024. Given these considerations we are not prepared to make these public here.

4- It is claimed that "SPHKAP and RI exhibited a strikingly punctate distribution in hippocampal pyramidal cell somata" (Line 79.). Since axons and dendrites are highly packed around soma in the hippocampus, it is hard to determine if the signals are from somata or from other compartments. Co-labeling with markers for soma, dendrites, and axon would make the point clear.

We thank the reviewer for this suggestion. We have now compared the spatial distribution of VGAT and GAD67 immunolabeling (markers of GABAergic terminals) with that of SPHKAP and Kv2.1, examining somata of neurons in hippocampal area CA1 in mouse brain sections (Supplementary Fig. 3a, b). Quantification of the SPHKAP's overlap with Kv2.1 compared to the colocalization of SPHKAP with GAD67 further supports the somatic localization of SPHKAP-type I PKA assemblies, which is also firmly established later in the manuscript by the SPHKAP immuno-EM results (Fig. 2i-k).

5- The super resolution imaging (Fig. 1f) is an adequate approach for analyzing the colocalization. However, the overlaps of the protein pairs could be resulted from coincidental co-localization. In particular, the overlap between Kv2.1 and SPHKAP, which is a novel finding, needs to be statistically analyzed to verify the results.

Thank you for this important comment. We have now performed extensive measurements and analyses of nearest neighbor distances of all protein pairs imaged using GSD super-resolution microscopy as shown in revised Fig. 2a-f. The results further support that SPHKAP is closely associated with Kv2.1, RI, and VAPs in hippocampal neurons.

6- The immunoEM analysis is potentially a key experiment to determine the localization of SPHKAP. However, the DAB enhanced signals in Fig. 1g are too

diffused for determining the actual location of SPHKAP. Also, the labeling with gold particles in Fig. 1h and 1i clearly shows that SPHKAP locates on the other side of ER facing the PM, as described in the diagram (Fig. 2h). These data do not align with the authors statement "type I PKA is concentrated at ER-PM junctions".

We agree with the reviewer's assessment that the EM data on SPHKAP and RI localization do not align with the use of our original term "ER-PM junctions". As such, as introduced above, we now refer to this as localization as being to "ER-PM junctional domains". We note that we have suspicions that the lack of immunogold labeling on the PM-facing ER membrane may be an artifact related to the strong fixation conditions required for EM, which when combined with the narrower gap between the PM and ER (10-15 nm) as compared to the space between subsequent stacked cisternae (20-30 nm) (e.g., see Fig. 2j and Siegesmund, *Anat Rec* **162**, 187-196 (1968)) could preclude antibody access to epitopes in this constrained space. That said, as we do not demonstrate that SPHKAP is within the ER-PM junction itself, we have modified the language of the manuscript throughout to clarify that it is associated with the ER cisternae associated with the ER-PM junctions within an "ER-PM junctional domain" rather than within the ER-PM junction itself.

7- The experiments using shRNA targeted to SPHKAP are critical experiments in determining the roles of SPHKAP. To exclude the possibility of off-target effects, multiple shRNA should be tested. Indeed, the authors showed there are three more efficient shRNA sequences (Extended data 8). For the analysis of PKA-C localization (Fig 3e.f), the authors should examined whether SPHKAP knock down affects the expression level of PKA-C.

We thank the reviewer for this important suggestion. We have now performed extensive additional experiments addressing this concern. These include PLA experiments (Fig. 2g, h; Fig. 5b), immunofluorescence labeling experiments (Supplementary Fig. 11d; Supplementary Fig. 12c), and live cell imaging experiments (Fig. 5c-f), using an additional distinct shRNA against SPHKAP. In total we analyzed 232 cells from 3 separate PLA experiments. Both anti-SPHKAP shRNAs used yielded similar results in all assays, supporting a specific effect of SPHKAP KD on the observed phenotypes. We have also now quantified overall PKA-C abundance (Supplementary Fig. 11d) in SPHKAP KD neurons and determined that it is unchanged relative to controls.

8- Their claim made in lines 175-178 is an overstatement.

We have modified the language to describe the overall protein complex as "PKA-regulated" rather than "PKA-dependent" (line 304).

9- It should be explained why the quantification of "depolarization-induced LTCC-dependent activation of somatic PKA in neurons" is an investigation for "whether

type I PKA localization near LTCCs impacts its function”.

Thank you for this important comment. We have now included additional language in the manuscript (lines 274-276) to highlight that there is strong precedence in the literature for AKAP-dependent anchoring of PKA near LTCCs and RyRs in neurons and cardiomyocytes, and that this PKA-Ca²⁺ channel signaling system is a key cellular strategy to regulate Ca²⁺ signals.

10- In Fig. 4C, the association of PKA-C with SPHKAP should be statistically analyzed.

We thank the reviewer for this suggestion. We have now included p-values from the statistical analysis of the PCC values of PKA-C:SPHKAP colocalization on what is now Fig. 6c of the revised manuscript.

Minor point

A scale bar for Fig. 1g (i) is probably incorrect.

We thank the reviewer for detecting this error. This has now been corrected.

Reviewer #2 (Remarks to the Author):

The manuscript by Vierra et al., 'ER-PM junctions couple excitation to Ca²⁺-1 activated PKA signaling' provide new insights into subcellular compartmentalization of soma-enriched type I PKA at ER-PM junctions. The authors identify SPHKAP (type I PKA anchors) as a protein interactor of KV2.1 channels. SPHKAP anchors type I PKA at ER-PM junctions and creates a PKA dependent signaling pathway to support Ca²⁺ influx and excitation-transcription coupling. Overall, the proposed model is very attractive and it shows for the first time a PKA signalosome associated to ER-PM, with an essential physiological role. However, some conclusions are overstated. Also, it is unclear how the formation of SPHKAP-RI condensates contributes to the functional role of ER-PM contacts in neurons. In addition, some experiments are not clearly explained in the main document. See below specific concerns.

We appreciate the reviewer noting that our manuscript “shows for the first time a PKA signalosome associated to ER-PM” junctional domains. We have carefully considered the important points raised by the reviewer and have addressed each of these with extensive new experiments and revisions to the text, resulting in an extensively revised and much improved manuscript. We thank the reviewer for their comments that we feel have substantively strengthened our manuscript and

provided additional support for our overall conclusions. We address each of the specific concerns below.

Major Points:

1. The authors show in Fig 1 and Fig S1 the subcellular localization of SPHKAP, Kv2 channels and RI subunit in mouse brain sections by confocal, TIRF and EM. Similar results in cultured hippocampal neurons (Fig S2). The conclusion of this data is that SPHKAP anchors type I PKA at ER-PM junctions. Although intensity profile lines showing overlap in specific regions, and correlation graph are presented (Fig 1C, Fig S1f), there is not quantitative evidence of how many of the Kv2 channel, SPHKAP and RI clusters overlap with each other. What is the percentage of Kv2 channel clusters associated to SPHKAP, RI and VAP? For instance, in Fig S2, many clusters of KV2.1 are observed, and only few are overlapping with RI and SHKAP. Data in Fig S2 should be quantified. The authors state that around a 47% of SHKAP is present in ER cistern stacks adjacent to PM, and 53% in ER cistern stacks no adjacent to PM. Is this consistent with number of SHKAP clusters associated to Kv2?

We thank the reviewer for suggesting that the spatial relationship of Kv2.1 channels with SPHKAP needs to be better quantified and have addressed this point with multiple new experiments. First, we have used super-resolution ground state depletion (GSD) microscopy of cultured hippocampal neurons to measure the spatial relationship of Kv2.1 with SPHKAP, as well as the relationship of SPHKAP with RI and VAPs, using the TIRF mode to assess proteins at/near the plasma membrane (Fig. 2a-f). From the reconstructed Kv2.1 and SPHKAP super-resolution maps we determined that $19.4 \pm 2.4\%$ of Kv2.1 pixels overlapped with SPHKAP immunolabeling – suggesting that approximately 20% of Kv2.1 is closer to SPHKAP than the 20-40 nm lateral resolution achievable using this microscopy technique. Similarly, measurement of nearest neighbor distances between SPHKAP and Kv2.1 (Fig. 2b) also indicated that about 17% of SPHKAP centroids were less than 50 nm from Kv2.1 centroids. We have described these results in the revised manuscript (lines 130-134).

We additionally performed proximity ligation assay (PLA) between Kv2.1 and SPHKAP in cultured hippocampal neurons (Fig. 2g) and found robust PLA signal in control neurons that is eliminated with SPHKAP knockdown, further supporting that Kv2.1 and SPHKAP molecules can be found close to each other (PLA detects protein pairs within 40 nm). Overall, these results indicate that Kv2.1 and SPHKAP clusters co-occur at a subset of ER-PM junctions, which can be appreciated from the conventional microscopy images of Kv2.1 and SPHKAP immunolabeling in neurons. Our data support that the large SPHKAP clusters obviously co-localized with Kv2.1 clusters in brain neurons (e.g., Fig. 1b, Supplementary Fig. 3) represent the subset of ER-PM junctional domains with stacked ER cisternae.

Regarding the fraction of SPHKAP-positive stacked ER cisternae found distal from the PM, we note that these structures have previously been observed by EM in brain sections (e.g., see Rosenbluth, *J Cell Bio* 1962, one of the original reports of ER-PM junctions/subsurface cisternae in brain neurons). We are unsure whether these ER stacks distal from the plasma membrane occur in vivo or are an artifact of sample preparation. For example, hypoxia, as can occur during perfusion, results in rapid (<1 min) dephosphorylation of Kv2.1, leading to its disconnection from ER VAPs and disassembly of the ER from the PM (Misonou et al., 2004; Fox et al., 2015; Kirmiz et al., 2018), also see Tao-Cheng et al., 2018. Thus, presence of these stacks deep in neurons in brain sections could represent previously PM-associated ER that has experienced acute sample preparation-dependent disconnection from Kv2.1. The absence of large SPHKAP clusters in Kv2 DKO neurons—which cannot form Kv2-type ER-PM junctions—suggests that this may be the case. Regardless, the presence of SPHKAP between these stacks further reinforces the apparent structural role of the SPHKAP-RI-VAP interaction in forming the stacked ER cistern structure.

2. In the presented data there is no evidence showing that SHKAP 'anchors' type I PKA at ER-PM (title section and Fig 1). Data shows co-distribution of Kv2, SHKAP and RI. Co-staining with VAP is also shown (TIRF, Fig. 1f; Fig S3). Here, it is not evaluated whether SHKAP can or not anchor type I PKA to ER-PM junctions. For this claim, the use of a shRNA against SHKAP or dominant negative to interfere with Kv2 should be provided.

We thank the reviewer for highlighting the imprecise language used to describe the spatial relationship of ER-associated SPHKAP-type I PKA assemblies with Kv2.1-associated ER-PM junctions. As described in our response to Reviewer 1 (prefatory remarks and remark #6), our data support that SPHKAP and type I PKA are present on the ER cisternae associated with ER-PM junctions, but we have not observed immunolabeling for these molecules within the junction itself. We have clarified this in the revised manuscript to highlight that SPHKAP-type I PKA is localized to stacked ER cisternae that are a component of the “ER-PM junctional domain” in neuronal somata rather than being detectably present within the ER-PM junction itself. While we have not attempted to investigate this further, we have suspicions that the lack of immunolabeling in the cytoplasmic cleft of the junction itself may be due to the inability of antibodies to access epitopes in the crowded ER-PM junction itself, as impeded by the harsh fixation conditions required for EM samples.

As recommended by the reviewer, we have also performed several new experiments using distinct anti-SPHKAP shRNAs, including PLA (Fig. 2g, h; Fig. 5b), immunofluorescence (Supplementary Fig. 11d; Supplementary Fig. 12c), and live cell imaging (Fig. 5c-f). We direct the Reviewer to our response to Reviewer 1, remark #2 for details. The results of these experiments further support our conclusion that SPHKAP-type I PKA assemblies are tightly associated with “ER-PM

junctional domains”, and specifically the ER cisternae associated with Kv2.1-dependent ER-PM junctions.

3. The role of Kv2 in SPHKAP clustering is also examined in Kv2 single and double KOs. Authors state ‘this suggest that Kv2 channels are determinant of SPHKAP-RI subcellular distribution but not expression’. Does Kv2 KO impair ER-PM contact or just clustering of SPKAP, RI? It looks like Kv2 DKO drastically impairs SHKAP clustering. Considering that around 53% of SHKAP present in stacked ER cisterns is not in contact with the PM, could Kv2 KO indirectly affect distribution of SHKAP? This should be discussed.

Thank you for this suggestion. We have modified the paragraph beginning at line 107 of the revised manuscript to emphasize that loss of ER-PM junction forming Kv2 channels correlates with a substantial reduction in SPHKAP cluster size. These results suggest that Kv2-dependent ER-PM junction formation promotes formation of large SPHKAP-type I PKA assemblies. The impact of Kv2 DKO on ER-PM junctions themselves is not known, however, clusters of ER ryanodine receptor are also significantly reduced in size (but otherwise persist) in brain sections from these mice (Kirmiz et al., 2018). This observation suggests that ER-PM junction are altered in neurons lacking Kv2 channels. We also appreciate the reviewer’s point about the large fraction of SPHKAP clusters found distal from the PM in our immuno-EM images. The absence of large SPHKAP clusters in Kv2 DKO neurons suggests that Kv2 channels play a role in the formation of SPHKAP-type I PKA assemblies in brain neurons. We note that large SPHKAP-type I PKA assemblies can form in the absence of Kv2 channels when expressed heterologously in HEK cells. We hypothesize that this difference may indicate a neuron-specific mechanism controlling SPHKAP-type I PKA assemblies that is dependent on Kv2 channels.

4. In Fig S3, VAPA/B co-distribution with SPHKAP and Kv2 is examined. Only representative images are shown. The authors state that VAP density co-varied with SPHKAP clusters. This is not clear from images and should be quantified in terms of number of clusters co-localizing with each other.

We thank the reviewer for this suggestion. We have included quantification of VAP enrichment at SPHKAP clusters in neurons measured in brain sections (Supplementary Fig. 3d) and in cultured neurons (Supplementary Fig. 7i, j).

5. Authors moved to heterologous expression in HEK cells to study SPHKAP-RI inter-dependency in cluster formation and to evaluate the role in ER remodeling (Fig 2, Fig S5). The clusters observed for mScarlet-tagged SPHKAP could be just an artifact of the tag aggregation at high expression levels. It is unclear whether images of co-expressed proteins (and deletion constructs) are examined at low or

high expression levels. Deletion constructs used are not clearly explained in main text. To confirm reciprocal co-clustering, the authors could use a smaller tag for SPHKAP (e.g., HA tag) However, it could be better to show the effect of shRNA for SPHKAP and RI, and rescue experiments with deletion constructs, or use a Dominant negative approach to specifically compete SPHKAP-RI association in cultured neurons. This will more clearly show the interdependency between them in cluster formation in neurons.

We appreciate the reviewer raising these important points. For analyses of protein localization in HEK cells and neurons, our imaging and analysis strategy relies on the acquisition of images from large numbers of cells, and the experimenter is often blinded to the identity of the sample. We acquired large, multi-tile images including several cells per field, and all cells with appreciable fluorescence or immunolabeling were included in analyses. As such, the data in Fig. 3 and 4 represents fluorescence measurements from over 800 individual cells and represents the full range of phenotypes (i.e., “low” to “high” expression) observed in our samples. Fig. 3b highlights the “low” and more rare “high” examples of SPHKAP expression and the impact on SPHKAP morphology – the relative cellular abundance of these two phenotypes can now be evaluated in Fig. 3d.

We have also better described the deletion constructs used in our studies in the revised manuscript. We thank the reviewer for suggesting that we assess the impact of fluorescent tags on SPHKAP-RI co-clustering. We have now included new data demonstrating that untagged SPHKAP and HA-tagged RI α display synergistic increases in cluster size similar to those seen with the fluorescent protein-tagged constructs (Supplementary Fig. 6a, b). These results further support that formation of large SPHKAP-RI co-clusters is intrinsic to their interaction and not an artifact of fluorescent protein expression.

To address the role of the endogenous SPHKAP-RI interaction in forming co-clusters in neurons, we assessed the impact of RIAD expression in cultured neurons. RIAD is a selective competitive inhibitor of the RI-AKAP interaction and thus precludes the association of RI with SPHKAP. We found that RIAD significantly reduced SPHKAP cluster size as compared to scrambled RIAD control (Supplementary Fig. 7a, b), supporting that the SPHKAP-RI interaction is required for formation of large SPHKAP clusters in neurons. We also note that GAD67⁺ neurons, which are presumably GABAergic interneurons, do not express SPHKAP and also do not display large clusters of RI or PKA-C. Quantification of interneuron PKA-C cluster size is presented in Fig. 6c.

6. The analysis of liquid-liquid phase separation properties of these SPHKAP-RI clusters is conceptually unclear. Is there any disordered region in these proteins that could suggest a role in maintaining phase separation of clusters in neurons? Hexanediol treatment has just little effect on SPHKAP-RI cluster formation. FRAP

experiments also show little mobility within assemblies. Droplet fusion is another property of phase separated assemblies, but this is not examined. The current data suggest that they are not liquid-liquid phase separated structures. The experiment with GFP, to study permeability of assemblies to molecules is not properly performed. Resolution of the experiment is not adequate to test this. At least FRAP for GFP should be performed to study GFP diffusion outside and within assemblies. The data does not add much understanding to the proposed model. Also, there are experiments in Fig S5, which are not explained, such as the use of forskolin/BMX, H89. I would suggest to remove all of this from the manuscript.

Thank you for this suggestion. However, we believe these experiments are important and should be included in the manuscript for the following reasons:

1. Two different models for intrinsic clustering of RI have been reported. The first study proposed a model that RI clustering depends on its physical association with AKAP11 (Day et al., *J Cell Bio* **193**, 347–363 (2011)), a SPHKAP paralog. The second model comes from a recent paper in *Cell* (Zhang et al., *Cell* **182**, 1531–1544 (2020)), which reported that PKA-RI α -GFP forms intracellular clusters through liquid-liquid phase separation. Given the similar appearance of SPHKAP-RI assemblies with the RI α -GFP condensates described in the *Cell* paper, we initially hypothesized that SPHKAP-RI assemblies would display similar liquid-like properties. However, we instead find that SPHKAP clusters are stable and do not appear to behave as “liquid” or “droplets,” at least by the same criteria/experiments used to evaluate RI α -GFP condensates in vivo. Moreover, when associated with SPHKAP, RI α displays the same non-liquid phenotype as SPHKAP. These immobile qualities may be related to the ER tethering of SPHKAP-RI assemblies. We also performed a preliminary assessment of SPHKAP cluster formation by performing time-lapse imaging in heterologous cells and did not observe “droplet”-like behavior. Given these two models for RI clustering, the recent intense overall interest in “liquid-liquid phase separation” and the recent highlighting of PKA-RI condensates as an example of this biological phenomenon, we believe it is important to share our data that when RI is associated with an AKAP, such as SPHKAP, it can acquire distinct non-LLPS qualities.
2. We have now performed additional FRAP experiments to demonstrate that SPHKAP-RI assemblies are permeable to molecules at least as large as GFP or DsRed-Monomer (Supplementary Fig. 9). This is an important point because although SPHKAP-RI assemblies are large and apparently immobile, they can conceivably allow signaling molecules such as PKA-C to diffuse in and out of the structure. It also demonstrates that these endogenous structures in neurons in the brain and in culture are not likely pathological,

impermeable “aggregates” but instead large and stable reservoirs for type I PKA near ER-PM junctions.

3. The disassembly of SPHKAP-RI assemblies in the presence of elevated cAMP is also a unique feature of the SPHKAP-type I PKA partnership in neurons and is completely opposite from the behavior of LLPS RI α -GFP condensates. The fact that forskolin/IBMX can cause SPHKAP-RI to decluster in the presence of H89 suggests that this effect is not due to PKA-C phosphorylation. We believe these data indicate a cAMP-sensitive mechanism for structural plasticity of stacked ER cisternae at ER-PM junctions in neurons. Presumably neurons can disassemble this structure under conditions of excessive cAMP signaling, perhaps to dampen PKA signaling.

7. The role of these ‘condensates’ (I would not call them condensates) in ER remodeling is not well examined. Distribution of VAP is examined in the presence of overexpressed SPHKAP-scarlet and RI-GFP in neurons (Fig S7). Expressed RI-GFP does not colocalize with VAPA/B clusters and total VAPA/B level appears to be reduced in this condition. SPHKAP-scarlet distribution is not clear, separated channels should have been provided not only for VAP. Compared to endogenous SPHKAP (Fig S2), the overexpressed protein label less/smaller SPHAP clusters. Changes in VAP looks modest, and perhaps a reduction in number and increase in size. Does clustering quantification account by these two parameters? Same experiment was performed in HEK cells (Fig 2g, Fig S7). In addition, EM of HEK cells co-expressing proteins increases OSER compared to non-transfected cells (Fig 2d). Are these OSER associated to the PM? Is expression of SPHKAP and mutants low or high expression? This is unclear. The conclusion that SPHKAP-RI assemblies remodel ER (title S7) is overstated, here only the distribution of VAP is examined. A clearer experiment to demonstrate a role of SPHKAP-RI complex in VAP distribution and ER remodeling would be knocking down SPHKAP and rescue with FL and mutants in cultured neurons and analyze VAP distribution and ER remodeling using a general ER marker such KDEL, Sec61b. Does SPHKAP KD impair stacked ER cisterns? Is ER-PM interaction impaired in this condition? These experiments could strengthen your proposed model presented in Fig 2h.

We thank the reviewer for raising these crucial points and for suggesting several excellent experiments to reinforce the statements we made in our original manuscript. We have addressed these concerns two ways. First, we revised the language in the manuscript to highlight that SPHKAP is a VAP-interacting protein rather than an ER-organizing protein. We have also revised the text to remove all mentions of SPHKAP-RI assemblies as “condensates”. We also determined that the images selected in the original manuscript did not effectively demonstrate the overall relationship of SPHKAP and VAPs. We have now provided in the revised manuscript extensive quantification of VAP immunolabeling and its relationship to SPHKAP in brain sections (Supplementary Fig. 3c, d) and neurons (Supplementary

Fig. 7i, j). We have also provided additional images of SPHKAP and VAP immunolabeling in HEK cells, along with new images of RI and PKA-C, as well as images that highlight the profound effect of SPHKAP FFAT mutants on the distribution of endogenous VAPs (Fig. 4b). We find that VAP immunolabeling in HEK cells is prominently co-clustered with WT SPHKAP-type I PKA assemblies, but any disruption of the SPHKAP FFAT motif completely abolishes this colocalization with VAPs (Fig. 4c). In the EM images of OSER structures in HEK cells we did not observe any particular association with the PM; however, it should be noted that these cells were not expressing Kv2.1 channels, and our data indicate that a subset of SPHKAP-RI structures can exist independently of the PM. EM analysis of ER structures in SPHKAP KD neurons would be an excellent experiment but was not technically feasible. We instead provide new experiments that examine the impact of treatment with cell penetrating TAT-FFAT peptides, which we designed to interfere with VAP-FFAT interactions. We found that treatment of neurons with these peptides uncoupled SPHKAP clusters from VAP and also caused disaggregation of VAP immunolabeling (Fig. 4e, f), suggesting that SPHKAP promotes the clustering of ER VAP proteins, presumably within stacked ER cisternae. Moreover, the robust PLA signal detected between SPHKAP and VAPs (Fig. 2h) in cultured neurons supports that SPHKAP and VAPs are tightly associated. We have revised the text throughout the manuscript to clarify that our data support that SPHKAP is a VAP-interacting protein rather than an ER-organizing protein.

8. It is not clear why the authors quantify distance between SPHKAP and PKA-C respect to Ca1.2 in Fig 3c-d. This is just confocal microscopy, there is not enough spatial resolution to examine the close proximity of these proteins. For this experiment better to show colocalization analysis or % of cluster colocalization. Although overlap between SPHKAP and PKA-C is clear, the co-distribution of SPHAP with KV2.1, Ca1.2 and RyR is not clear. PLA is a nice way to study the interaction at the nanoscale resolution, but why there are some puncta outside the soma? Is SPHAP distributed also along dendrites? (Fig 3e). Since the authors have a good shRNA against SPHKAP, it would be good to study the requirement of SPHKAP in the distribution of markers shown in Fig 3A. Is SPKAP playing a structural role in connecting ER-PM to ca2+ channels and PKA signaling? Or there is only a functional role as shown in Fig 3g-k?

We thank the reviewer for these comments and suggestions. To address these points, we have included additional language in the manuscript (lines 274-276) as also described to Reviewer 1 (remark #9). We have also performed additional PLA experiments between RyR and RI to assess whether SPHKAP is placing type I PKA near RyR ER Ca²⁺ channels (Fig. 5b). We find robust RyR-RI PLA in control neurons, but not in neurons expressing different anti-SPHKAP shRNAs, further supporting that SPHKAP is placing type I PKA close to Ca²⁺ channels at ER-PM junctional domains. With regard to additional PLA signal outside the soma, we suspect that

this could reflect a minority population of SPHKAP in dendrites. We have not assessed the characteristics or functions of this relatively minor population of SPHKAP in this manuscript, which focuses on the much more prominent SPHKAP-RI assemblies localized in the soma. Future studies may investigate whether there are roles for these minor populations of SPHKAP outside of the soma, but we believe it is beyond the scope of the present study.

The reviewer raises an interesting point about a possible structural role for SPHKAP in organizing Ca^{2+} channels at these ER-PM junctional domains. At this point our data cannot differentiate the relative importance of the signaling versus structural roles of SPHKAP-type I PKA in mediating somatic Ca^{2+} signaling. We note that we now have performed additional Ca^{2+} imaging experiments using multiple anti-SPHKAP shRNAs and find that while SPHKAP knockdown eliminates the L-type Ca^{2+} channel-mediated component of this response, the nimodipine-insensitive component of Ca^{2+} influx (i.e., the non L-type Ca^{2+} channel fraction) is unaffected by SPHKAP KD. These new data are included in the revised manuscript in a revised Figure 5. The L-type Ca^{2+} channel-mediated component supported by SPHKAP may include LTCC-triggered CICR. It is interesting to speculate that stacked ER cisternae may amplify RyR-dependent CICR by providing a larger overall reservoir of luminal Ca^{2+} and allowing for horizontal apposition of RyRs in adjacent ER membranes. Future studies will also examine this possibility.

Minor comments:

1. Figure 2a is not clearly explained in result section.

Thank you for this comment. The constructs (now illustrated in Fig. 3a) are now clearly described in lines 171-177 of the revised manuscript.

2. In extended Figure 5 and 6, authors are checking whether SPHKAP clusters have LLPS properties. There is not reference to Forskolin/ IBMX and H89 experiment.

We thank the reviewer for pointing out this oversight. We now describe the rationale for the Forskolin/IBMX experiments in lines 213-217 of the revised manuscript.

3. There are no scale bars for extended figure 5d

We have corrected this error and included a scale bar on Supplementary Fig. 7d of the revised manuscript.

4. It should be explained why PLA assay was used in the result section

We now describe the rationale for the PLA experiments in lines 135-136 of the revised manuscript.

Reviewer #3 (Remarks to the Author):

This is a substantial investigation that includes a number of interesting and novel observations. Notably, the authors show that: (i) SPHKAP-PKA type I associates with Kv2.1 channels resulting in localisation of the complex at ER-PM junctions. This part of the study is robust and very firmly established using multiple complementary techniques including beautiful EM imaging (ii) They show that interaction between PKA RI subunits and SPHKAP supports clustering of both proteins and is linked to generation of organised smooth ER (iii) They show that SPHKAP is required for Ca²⁺ elevation triggered by potassium-induced activation of voltage-gated calcium channels (iv) they show that SPHKAP is necessary for activation of PKA that results from Ca²⁺ elevation following activation of these channels.

Overall, this is an impressive body of work, and will interest researchers in EM-PM junctions, and in localised cAMP/PKA signalling. Nevertheless, I was not convinced of the logic of the mechanism presented in Fig. 2h, and there are also some uncertainties regarding exactly how SPHKAP is supporting PKA activation following activation of LTCCs (Figure 4).

We are pleased that the reviewer found our manuscript to be “an impressive body of work, and will interest researchers in EM-PM junctions, and in localised cAMP/PKA signalling.” We have substantively addressed the reviewer’s comments and concerns with extensive new experiments and revisions to the text of the manuscript that we feel have greatly strengthened our study and provide further support for our conclusions.

Specific comments:

1. With respect to the relative importance of different PKA regulatory subunit isoforms. It would be worth comparing their findings with studies that have imaged PKA subunit distribution in the hippocampus (Ilouz et al., eLife, 2017) and quantified the abundance of PKA subunits in different brain regions (Walker-Gray, PNAS, 2017). Potentially, Ilouz et al., indicates that CA1 neurons are somewhat unusual in expressing R1beta at high levels in the cell body.

We thank the reviewer for pointing out this oversight. We have now described the important findings of Ilouz et al. and Walker-Gray et al. in the context of our observations in lines 91-93 and 99-102 of the revised manuscript.

2. Furthermore – as summarised by Kirschner et al., Mouse models of altered protein kinase A signaling – there is evidence that R1beta is particularly important

in the hippocampus

Thank you for directing us to this informative review, which reminded us of the interesting results described by Brandon et al. (1995). We have now described these significant prior observations in the context of our results in lines 379-381 of the revised manuscript.

3. Line 3-76. SPHKAP distribution is exclusively somatic rather than somatodendritic judging by the supplementary images

We appreciate this comment and agree that SPHKAP immunolabeling is more somatically restricted than Kv2.1 in CA1 pyramidal neurons, however rare puncta can be detected in proximal dendrites of these cells (e.g., see Supplementary Fig. 3). Interestingly, dispersal of SPHKAP into dendrites is apparent in hippocampal area CA2 – a region where Kv2.1 expression is substantially less than in CA1 (Palacio et al., 2017). SPHKAP immunolabeling can also be detected in the dentate gyrus molecular layer which is where the dendrites of dentate granule cells are located (Supplementary Fig. 1a). The RI immunolabeling pattern appears to generally follow that of SPHKAP in these hippocampal regions.

4. In figure 2, and on lines 116-129, it could be emphasised that – unusually – SPHKAP has two RI anchoring sites. So it is immediately apparent how SPHKAP could support RI clustering (but not so the converse)

We thank the reviewer for raising this key point. We have now emphasized this unique property of SPHKAP (and its paralog, AKAP11) in lines 168-170 and 387-390 of the revised manuscript. We also performed additional experiments to evaluate the role of RI anchoring in affecting the clustering of these AKAPs (e.g., Fig. 2f, Supplementary Fig. 6d, e). Our results demonstrate that RI anchoring to SPHKAP and also AKAP11 is intimately linked to formation of large AKAP-RI co-clusters. As discussed further below we are still unsure of the specific molecular mechanism.

5. I was confused as to how figures 2c and extended figure 5a relate – the numbers are different whereas the conditions look the same?

We apologize for the lack of clarity here. The frequency distribution presented in Fig. 2c (now Fig. 3c of the revised manuscript) is generated from the pooled SPHKAP cluster sizes from all analyzed cells (i.e., nearly 2000 SPHKAP puncta total). The individual values now reported in Fig. 3d and Supplementary Fig. 6c (previously extended Fig. 5a) are the mean values obtained from single cells. We have ensured that what is being measured and presented is more clearly stated in the figure legends.

6. It is not clear why the D/D-linker version of RIalpha is able to support clustering of SPHKAP... Zhang et al., Cell, 2020 (paper on RI phase separation) clearly shows that the cyclic nucleotide binding domains of RI are required for phase separation of RI. How do the authors reconcile these observations?

We thank the reviewer for this comment. Based on the results of Zhang et al. we initially hypothesized the RI LLPS was responsible for formation of the large SPHKAP-RI assemblies. The finding that the D/D+linker domain could drive clustering was thus unexpected and we have been unable to deduce the molecular mechanism of how RI association with SPHKAP drives formation of large co-clusters. To address the reviewer's remark, we have performed new experiments to further investigate the domains on RI necessary for SPHKAP clustering. We have now obtained the D/D domain plasmid from the Zhang lab and in new experiments using this key reagent found that the D/D domain by itself is sufficient to promote formation of these large clusters (Fig. 2b-d). Because the necessary domains required for RI LLPS (i.e., the disordered linker and the CNB domains) are presumably absent in the D/D-only construct, our results suggest that RI itself is not driving phase separation of SPHKAP-RI co-clusters, but that these clusters arise via a distinct mechanism. Our FRAP and hexanediol experiments also suggest that this mechanism is not through the formation of LLPS-generated condensates. However, RI anchoring to SPHKAP (as well as AKAP11) does appear to be necessary, as RIAD completely abolishes large clusters of both SPHKAP (Fig. 2f) and AKAP11 (Supplementary Fig. 6d, e, and Day et al., *J Cell Bio* **193**, 347–363 (2011)). Moreover, VAP association/ER tethering appears to be dispensable for formation of SPHKAP-RI assemblies, as these structures persist in FFAT mutants (Fig. 4b) or when the SPHKAP-VAP interaction is disrupted via introduction of TAT-FFAT interfering peptides (Fig. 4e).

7. Related to this point, in the summary figure presented in fig. 2h, in the cartoon RI is presented as bridging two SPHKAP molecules. This does not seem logical given what is known about the structural basis of RI-SPHKAP interaction. Means et al., 2011 showed that RI binds to amphipathic anchoring helices presented by SPHKAP via the RI dimerization and docking domain – ie there is no way for an RI dimer to bridge two copies of SPHKAP given that dimerization and anchoring to a single site occur within the same small domain. Potentially phase separation-type interaction of RI dimers bound to different SPHKAP polypeptides could support SPHKAP clustering, However, the hexane diol experiments presented in extended figure 5 do not support an important role for phase separation in the clustering process. I thought this was the least convincing and most confusing aspect of the study.

We appreciate the reviewer's confusion with these results and unfortunately cannot yet provide a detailed molecular explanation for how RI association with SPHKAP promotes the formation of the large SPHKAP-RI coclusters. However, our results including from extensive new experiments performed in response to the reviewer's concerns demonstrate that RI anchoring to SPHKAP is necessary, and we have now

acquired new single cell measurements from over 1000 cells to address this point. We have now found that RIAD, a specific inhibitor of RI-AKAP interaction, precludes formation of large SPHKAP-RI clusters in HEK cells (Fig. 3f) and de-clusters endogenous SPHKAP in neurons (Supplementary Fig. 7a, b). Moreover, our new data indicate that the responsible element of RI lies within the D/D domain, as this fragment by itself was sufficient to drive the formation of large coclusters when expressed with SPHKAP.

Interestingly, the D/D fragment of RI (aa 12-61) contains an additional domain that can interact with the "RI specifier region" (RISR) present in some AKAPs that has previously been shown to mediate interactions outside of the amphipathic PKA binding domain (Jarnaess et al., 2008). It would be interesting to determine if a similar RISR(s) is present in SPHKAP and if so, if it enabled an RI dimer to interact with two separate AKAPs. Because each SPHKAP (and AKAP11) molecule can accommodate two type I holoenzymes, such a model might account for the formation of large, RI-crosslinked SPHKAP-RI clusters.

We have made substantial efforts to clarify the rationale for these experiments in the revised manuscript and list why we think these data are relevant and important to include in our response to Reviewer 2, remark 6.

8. In figure 3c & d – what is the average distance to a randomly selected point in the soma? (ie the average distance will naturally reduce as the density of SPHKAP/PKA-C points increases)

Thank you for this comment. We have performed several new experiments that we hope will address this remark. First, we performed measurements of nearest neighbor distances of SPHKAP with VAPs in GSD super-resolution images of cultured neuronal somas (Fig. 2f). Note that VAPs have a much broader cellular distribution compared to the somatic enrichment of SPHKAP such that a "randomly selected point in the soma" would likely be close to a point of VAP immunolabeling. When measuring distance of SPHKAP points to VAP points, we find there is an apparent bimodal distribution in distance: 1) a large population that is very close to SPHKAP (presumably these are the VAP molecules interacting with SPHKAP); and 2) what is apparently the remainder of the VAP population, which shows a much broader distribution in distances from SPHKAP.

Second, we performed PLA in cultured neurons to define populations with proximity to one another of SPHKAP and Kv2.1 (Fig. 2g) and RyRs and RI (Fig. 5b). Control neurons showed robust PLA between both pairs of proteins which was abolished in SPHKAP KD neurons. The somatic distribution of the PLA signal in the representative images provides a sense of where SPHKAP is closely associated (i.e., within 40 nm to generate a PLA signal) with Kv2.1 and also, given the association of SPHKAP with RI, with RyRs.

9. Regarding reference 33 (Dell'Acqua study) – the authors state that 'consistent with the observation that direct inhibition of PKA or its uncoupling from AKAPs suppresses LTCC activity...' I feel this is a little misleading since the work from Dell'Acqua/Sather highlights a critical role for AKAP79-type II PKA complexes. Ie, the notion that SPHKAP-type I PKA is key to PKA activation downstream of LTCCs challenges these studies rather than being consistent with them. The authors should acknowledge this difference and comment on how they view the relative importance of these two AKAPs in responding to Ca²⁺ influx through LTCCs.

Thank you for this comment. We feel our work does not conflict with the work of the Dell'Acqua and Sather labs, which rigorously demonstrates the sensitivity of neuronal LTCCs to regulation by AKAP79-anchored PKA through a variety of complementary approaches. The primary rationale for this statement comes from a key experiment described in Dittmer et al., *Cell Rep* **7** (2014). In Fig. 2b of this study the authors demonstrate that Ht31 acutely introduced via the patch pipette leads to reduced LTCC currents with an IC₅₀ of 8.9 μM. At this concentration, Ht31 would be predicted to be displacing both RI and RII from AKAPs, as the K_d for the RIα-Ht31 interaction was calculated to be 2.1 μM (Burton et al., *PNAS* 1997). Given the abundance of SPHKAP-RI complexes in the soma and their proximity to LTCCs, we think it is reasonable to predict that a part of the Ht31-sensitive LTCC current measured in this experiment was carried by LTCCs near SPHKAP-RI assemblies.

We appreciate the suggestion to highlight the differences between the SPHKAP-type I PKA system and type II PKA systems of distal dendrites. We have now commented on the distinct qualities of the somatic SPHKAP-type I PKA system as compared to the type II system (e.g., AKAP79/150) in lines 370-374 of the revised manuscript.

*10. In terms of interpreting the results of figure 4 – this is an interesting finding although some uncertainties remain around the mechanism. Specifically:
(i) Is Ca²⁺ release through ryanodine receptors required for the elevation in Ca²⁺ seen in e.g. Fig 3g?*

We thank the reviewer for this question. We have now performed additional Ca²⁺ imaging experiments to better understand how SPHKAP KD causes reduced depolarization-induced Ca²⁺ influx (Fig. 5c-f), performing measurements in over 300 additional control and SPHKAP KD neurons. We included in these experiments a second distinct anti-SPHKAP shRNA to confirm that the observed effects were due to SPHKAP KD and not an off-target effect of the shRNA. We chose to examine how SPHKAP KD impacted the LTCC-sensitive component of Ca²⁺ influx. We find that while SPHKAP KD dramatically reduces the depolarization-triggered Ca²⁺ signal, this treatment selectively eliminates the LTCC (nimodipine-sensitive component, while

the nimodipine-insensitive component of Ca²⁺ influx is unaffected (Fig. 5d, f). These results indicate that SPHKAP is specifically affecting the component of the depolarization triggered Ca²⁺ signal contributed by LTCCs.

We also determined that SPHKAP is placing type I PKA near RyRs, as PLA signal between RyRs and RI was abolished by SPHKAP KD (Fig. 5b). Thus, we speculate that RyR function could be impacted by SPHKAP-anchored PKA as well. Future experiments will determine the relative participation of RyRs to the SPHKAP-sensitive Ca²⁺ signal.

(ii) Do the authors think that C subunits binds to SPHKAP-RI are essential for the PKA activity measured in the experiments presented in figure 4? Note, they detected robust C subunit co-precipitation with Kv2.1 channels by MS. Conceivably, however, the complex could serve a purely structural role given that SPHKAP knockdown prevents LTCC-triggered Ca²⁺ elevation (Fig 3g).

Regarding these points, ideally the authors could perform additional experiments with RyR blockers, and potentially using a knockdown-replacement approach to substitute in mutated versions of RI subunits that are not able to interact with PKA C subunits. If this is not feasible then they should more clearly highlight remaining uncertainties in the mechanism.

Thank you for this comment. Note that we also observed robust PKA-C co-clustering with SPHKAP and RI in neurons (e.g., see Supplementary Fig. 2). Because PKA-C tends to disperse from SPHKAP clusters during depolarization (Fig. 6c, note the drop in PCC between SPHKAP and PKA-C), we believe this is supportive of our model that PKA-C associated with SPHKAP-RI is participating in the measured depolarization-induced increase in PKA activity. We additionally have now measured forskolin/IBMX PKA activation in SPHKAP KD neurons (Supplementary Fig. 12d) and found that it did not differ from control neurons, suggesting that while SPHKAP anchoring of type I PKA supports depolarization-induced activation of PKA, it does not contribute to PKA-C activity in response to forskolin/IBMX treatment.

The suggested experiments using RyR blockers and knockdown replacement are excellent ideas but were not feasible for the present study. We cannot rule out a role for the apparent structural function of SPHKAP-type I PKA at the stacked ER cisternae, and future experiments will seek to address this through dissociating the structural function from the signaling role in anchoring of PKA-C at these sites.

** See Nature Portfolio's author and referees' website at www.nature.com/authors for information about policies, services and author benefits.

REVIEWERS' COMMENTS

Reviewer #1 (Remarks to the Author):

The authors have addressed most of my concerns and the logic of the manuscript is now much easier to follow.

One last thing this reviewer is still not sure is the authors' interpretation about the localization of SPHKAP. In particular, the result of the proximity labeling analysis between Kv2.1 and SPHKAP is not reflected in the model shown in Fig. 4. Interpretation of the PLA result is better to be incorporated if the authors claim that SPHKAP is still in the 40 nm proximity to each other.

Also, I guess the scale bar still seems to be wrong.

Reviewer #2 (Remarks to the Author):

The manuscript by Vierra et al., has enormously improved after revision. The authors have performed new experiments addressing the concerns. The statements/conclusions fit the presented data.

I have one additional suggestion, which I hope the authors can easily add to their manuscript.

Considering the proposed model in Fig 4g, the EM data (Fig 2 i-k), and the new PLA data showing SPHKAP-Kv2.1 interaction (Fig. 2g), it is difficult to understand how SPHKAP present in between stacked ER, but not facing PM, could give a signal for PLA (40nm max distance for signal). For this, consider the distance between ER-PM, width of ER lumen, and the barrier for the interaction of these proteins, the ER membrane facing the PM. All the controls are present in the PLA experiment, indicating this is a real interaction. In the response to reviewers, the authors mention that it is possible that SPHKAP also distributes along the ER facing the PM, but perhaps there is a technical issue to visualize this with EM. Could this be shortly explained in result or discussion section? this will help the reader to understand the possible contrasting results showed in these Figs.

Reviewer #3 (Remarks to the Author):

The authors have effectively addressed my comments. Congratulations on an interesting study.

REVIEWERS' COMMENTS

We thank all three reviewers for reading our revised manuscript. We greatly appreciate their comments, which have improved the rigor the study. We address the remaining minor comments below:

Reviewer #1 (Remarks to the Author):

The authors have addressed most of my concerns and the logic of the manuscript is now much easier to follow.

One last thing this reviewer is still not sure is the authors' interpretation about the localization of SPHKAP. In particular, the result of the proximity labeling analysis between Kv2.1 and SPHKAP is not reflected in the model shown in Fig. 4.

Interpretation of the PLA result is better to be incorporated if the authors claim that SPHKAP is still in the 40 nm proximity to each other.

Also, I guess the scale bar still seems to be wrong.

We appreciate that the reviewer found that the revisions have improved the overall manuscript. We also thank the reviewer for their suggestion that the illustrated model be updated to incorporate the results of the PLA experiments. We have now done this, presented in Fig. 5g of the revised manuscript.

We also apologize for failing to incorporate the corrected scale bar of the noted figure panel (now Fig. 3a(i)). We inadvertently used the original (incorrect) version of this figure when submitting the revised version of the manuscript. It should now be accurate.

Reviewer #2 (Remarks to the Author):

The manuscript by Vierra et al., has enormously improved after revision. The authors have performed new experiments addressing the concerns. The statements/conclusions fit the presented data.

I have one additional suggestion, which I hope the authors can easily add to their manuscript.

Considering the proposed model in Fig 4g, the EM data (Fig 2 i-k), and the new PLA data showing SPHKAP-Kv2.1 interaction (Fig. 2g), it is difficult to understand how SPHKAP present in between stacked ER, but not facing PM, could give a signal for PLA (40nm max distance for signal). For this, consider the distance between ER-PM, width of ER lumen, and the barrier for the interaction of these proteins, the ER membrane facing the PM. All the controls are present in the PLA experiment, indicating this is a real interaction. In the response to reviewers, the authors mention that it is possible that SPHKAP also distributes along the ER facing the PM,

but perhaps there is a technical issue to visualize this with EM. Could this be shortly explained in result or discussion section? this will help the reader to understand the possible contrasting results showed in these Figs.

We thank the reviewer for the positive feedback on the revised manuscript. We also appreciate their excellent points about the interpretation of the PLA experiment and suggestion to update the model accordingly. We have now revised the illustrated model (now Fig. 5g) and have also included additional discussion in the manuscript (lines 161-171) as suggested by the reviewer.

Reviewer #3 (Remarks to the Author):

The authors have effectively addressed my comments. Congratulations on an interesting study.

We thank the reviewer for their suggestions which strengthened the manuscript.